



# An evaluation of the Arabian Sea Mini Warm Pool's advancement during its mature phase using a coupled atmosphere-ocean numerical model

Sankar Prasad Lahiri[1], Kumar Ravi Prakash[1,2], Vimlesh Pant[1]

5  [1]Centre for Atmospheric Sciences, Indian Institute of Technology Delhi, India.

[2]Applied Physics Laboratory, University of Washington Seattle, WA, USA.

*Correspondence to*: Sankar Prasad Lahiri (sankarprasadlahiri2@gmail.com)

**Abstract.** A coupled atmosphere-ocean numerical model has been used to examine the relative contributions of atmospheric and oceanic processes in developing the Arabian Sea Mini Warm Pool (MWP). The model simulations were performed for three independent years, 2013, 2016, and 2018, through April-June, and the results were compared against observations. The model simulated sea surface temperature (SST) and salinity bias were less than 1.75ºC and 1 psu, respectively; this bias was minimal in the MWP region. Moreover, the model simulated results effectively represented the presence of the MWP across the three years. The mixed layer heat budget analysis indicates that the net heat flux raised the mixed layer temperature tendency of the MWP by a maximum of 0.1°C/day during its development phase. The vertical processes, thereafter, exerted a cooling impact of -0.08ºC/day in the temperature tendency, causing it to dissipate. Further, four sensitivity numerical experiments were performed to investigate the comparative consequences of the ocean and atmosphere on the advancement of the MWP. The sensitivity experiments indicated that pre-April ocean conditions in years with a strong MWP result in a 136% increase in MWP intensity in years when MWP SST was close to climatology, which shows the primary role of oceanic preconditioning in determining MWP strength during strong MWP years rather than the atmospheric forcing. However, once the oceanic preconditions are met, the atmospheric conditions of weak MWP years lead to an 82% reduction in MWP intensity relative to normal years, highlighting the detrimental impact of atmospheric forcing under such circumstances. Additionally, atmospheric conditions, particularly wind, are critical in influencing the spatial evolution and dissipation of MWP in the SEAS by modulating vertical processes. A wind shadow zone, characterized by less turbulent kinetic energy that does not exist during weak MWP years, facilitates the spatial expansion of MWP in SEAS during moderate to strong MWP years.



# 1 Introduction

The North Indian Ocean's (NIO) associated ocean-atmosphere dynamics, including monsoon and cyclones, are well explored by researchers. One of the primary determinants in this interconnected process is sea surface temperature (SST). During the pre-monsoon season, the southern region of the Arabian Sea experienced SST exceeding 28°C, which is associated with the larger Indo-Pacific Warm Pool. However, the highest temperature has been observed in the southeastern Arabian Sea (SEAS) from late April to May, before the onset of the Indian Summer Monsoon. These patches of warm water in the SEAS are often referred to as the Arabian Sea Mini Warm Pool (MWP) (PV Joseph, 1990; Rao & Sivakumar, 1999; Seetaramayya & Master, 1984; Shenoi et al., 1999). The MWP SST remains 0.5°C to 1°C higher than the surroundings during this time. Because the MWP SST stays above 30°C, well above the minimal criteria for deep convection, it is thought to play a significant role in the Indian Summer Monsoon characteristics over Kerala (Deepa et al., 2007; Masson et al., 2005; Neema et al., 2012; R. R. Rao & Sivakumar, 1999).

Extensive studies have been conducted on the seasonal and interannual evolution of the MWP due to its noteworthy impact on regional climate dynamics (Akhil et al., 2023; Durand et al., 2004, 2007; Kurian & Vinayachandran, 2007; Mathew et al., 2018; Nyadjro et al., 2012; Rao & Sivakumar, 1999; Shenoi et al., 1999). According to Rao and Sivakumar (1999), the East India Coastal Current brings low salinity water to the SEAS in winter, which causes strong stratification and leads to the formation of the MWP in May. Shankar & Shetye (1997) explained the formation of the MWP in terms of the wave propagation phenomenon. The coastal kelvin wave travels along India's east coast after the summer monsoon has passed, eventually reaching the SEAS in November – December. Later, it is deflected westward by the Rossby Wave, forming the 'Laccadive High' (Bruce et al., 1994, 1998; Shankar & Shetye, 1997). The East India Coastal Current, triggered by the coastal kelvin wave on India's east coast, brings low salinity water to the SEAS and recirculates along the downwelling Laccadive High eddy in November and December. This low salinity water leads to the formation of a barrier layer in the SEAS (Gopalakrishna et al., 2005; Kumar et al., 2009; Masson et al., 2005; Murty et al., 2006; Shenoi et al., 1999). The barrier layer prevents the mixing of water above and below the thermocline (Lukas & Lindstrom, 1991) and traps the incoming shortwave radiation during the pre-monsoon season within the top few meters and thus increases the SST (Hastenrath & Greischar, 1989). Masson et al. (2005) reported that the wintertime barrier layer intensifies the SST of the MWP by approximately 0.5°C.

Kumar et al. (2009) investigated the development of MWP using the Princeton Ocean Model. They stated that salinity has a significant impact on the formation of MWP. Using an ocean general circulation numerical model, Kurian & Vinayachandran (2007) discovered that the orographic influence of the western ghat acts as a wind barrier, increasing net heat flux and, hence, MWP intensity. According to Mathew et al. (2018), latent heat flux and incoming shortwave radiation, rather than wintertime freshening, influence the formation of MWP. Recently, Akhil et al. (2023) found that subsurface dynamics during the preceding winter have very little influence on the development of MWP.

Warm water can negatively affect the ocean ecosystem, particularly coral reefs (Abram et al., 2003; Doval & Hansell, 2000; Sarma, 2006). Given its proximity, any sudden increase in the intensity of the MWP could impact biological activity in





the Laccadive High region. The MWP's impact also affects sound propagation dynamics (Kumar et al., 2007). Despite the
evident importance, the formation mechanism of MWP remains a topic of ongoing scientific debate. The complex interplay of
factors contributing to the genesis of the MWP, including winter salinity stratification and the presence of the Western Ghats,
has been investigated in previous studies (Durand et al., 2004; Gopalakrishna et al., 2005; Kumar et al., 2009; Kurian &
Vinayachandran, 2007; Masson et al., 2005; Nyadjro et al., 2012). However, recent findings by Akhil et al. (2023) suggested
that the influence of winter salinity stratification on the MWP genesis might be less significant than previously thought.
Moreover, a limited number of studies have focused on the air-sea interaction during the mature phase of the MWP,
highlighting the need for further research in this area. Kumar et al. (2009) and Mathew et al. (2018) have explored these
interactions, but comprehensive analyses remain sparse. Li et al. (2023) recently examined the Arabian Sea warm pool during
its mature phase, emphasizing its seasonal and interannual variability using reanalysis datasets. While these studies contribute
valuable insights, they primarily address variability over time rather than the specific processes driving the MWP's
development and dissipation.

In this study, we examine the impact of the air-sea interaction on the progression of the Arabian Sea Mini Warm Pool
using a coupled atmosphere-ocean numerical model. The coupled model employed in this study is calibrated for precise
application on a seasonal time scale, which imposes limitations on the duration of simulations. Therefore, long-term
simulations in each year are not feasible. Consequently, the present study focuses on configuring a regional coupled
atmosphere-ocean numerical model to study the region-specific expansion and dissipation of the MWP. We also aim to
elucidate the contribution of oceanic and atmospheric conditions to its growth over the years with distinct MWP intensity. The
primary framework of the study is as follows: Section 2 provides an in-depth discussion of the data and methodology, including
a detailed explanation of the coupled numerical model. Section 3 demonstrates the results along with the coupled model's
ability to simulate temperature, salinity, and currents. Further, a few model sensitivity experiments have been incorporated to
explore MWP characteristics and the influence of atmospheric and oceanic conditions across three different MWP events.
Section 4 discusses the results and concludes the study with key findings.

## 2    Data & Methodology

### 2.1    Model Details

The coupled Atmosphere-Ocean model used in this study consists of the atmospheric model 'Advanced Research
Weather Research and Forecasting' (WRF – ARW) and the ocean model 'Regional Ocean Modeling System' (ROMS). Both
models and the model coupling toolkit (MCT) are part of the Coupled Ocean-Atmosphere-Wave-Sediment Transport Modeling
System (COAWST) (Warner et al., 2010). The MCT was used to couple the Atmospheric model WRF-ARW and ocean model
ROMS (Jacob et al., 2005). This coupled numerical model was used to study the air-sea interaction during tropical cyclones
(Chakraborty et al., 2022; Prakash et al., 2018; Prakash & Pant, 2017, 2020; Zambon et al., 2014), coastal processes (Carniel



et al., 2016; Olabarrieta et al., 2011, 2012; Ricchi et al., 2016; Kumar & Nair, 2015) and monsoon deep depression over the
Bay of Bengal(Chakraborty et al., 2023).

The ROMS model is a free-surface, hydrostatic, three-dimensional primitive equations (i.e., Reynolds averaged
Navier-Stokes' equation) ocean model widely used in estuarine, coastal, and basin-scale research. The primitive equations in
the ROMS model are solved using boundary-fit orthogonal, curvilinear coordinates on a staggered Arakawa C grid (Arakawa
& Lamb, 1977). This model employs a terrain-following vertical sigma-coordinate system (Haidvogel et al., 2000; Phillips,
1957; Song & Haidvogel, 1994). The ROMS model has been widely used to investigate the coastal, open ocean, and
biogeochemical processes(Nigam et al., 2018; Paul et al., 2023; Sandeep et al., 2018; Seelanki et al., 2021).

The WRF - ARW atmospheric model is a non-hydrostatic, fully compressible model that predicts mesoscale and
microscale processes using boundary layer physics schemes and various physical parameterizations (Skamarock & Klemp,
2008; Skamarock, 2008). On a horizontal Arakawa C grid and a vertical sigma-pressure coordinate, WRF - ARW estimates
wind momentum components, surface pressure, longwave and shortwave radiative fluxes, dew point, precipitation, surface
sensible and latent heat fluxes, relative humidity, and air temperature. WRF is intended to serve and support atmospheric
research and operational forecasting needs (Dai et al., 2013). Parameterization schemes are available in microphysics, cumulus
parameterization, planetary boundary layer, surface layer, land surface, and longwave and shortwave radiations, with multiple
options for each process. In the COAWST modeling system, the WRF code has been modified to provide improved bottom
roughness when computing bottom stress over the ocean (Warner et al., 2010). The components of the wave and sediment
transport models are not used in this study.

## 2.2    Model Configuration and Experiment Design

A coupled atmosphere-ocean (WRF+ROMS) model was configured over the north Indian Ocean (Fig. 1) to study the
influence of oceanic and atmospheric variables in the progress of MWP. WRF supports a variety of parameterization schemes.
In our WRF configuration we have incorporated the WRF Single-moment-6-class Scheme (Lim & Hong, 2010) as
microphysics parameterization to represent grid-scale precipitation processes, the New Tiedtke Scheme for cumulus
parameterization that illustrates sub-grid scale convection and cloud detrainment (Zhang & Wang, 2017), the Yonsei
University Scheme (Hong et al., 2006) for planetary boundary layer physics, and the MM5 Similarity Scheme (Paulson, 1970).
We have used the Unified Noah Land Surface Model (Tewari et al., 2004). At each time step, the atmospheric and land surface
models calculate exchange coefficients and surface fluxes of the land or ocean layer and send them to the Yonsei University
planetary boundary layer (Hong et al., 2006). The Dudhia Shortwave Scheme (Dudhia, 1989) is used for shortwave radiation
parameterization, and the RRTM Longwave Scheme (Mlawer et al., 1997) is used for longwave radiation parameterization.
ROMS, the ocean model, employs several numerical schemes to describe sub-grid processes accurately. Vertical mixing is
handled by the K-profile parameterization (KPP) mixing scheme (Large et al., 1994).



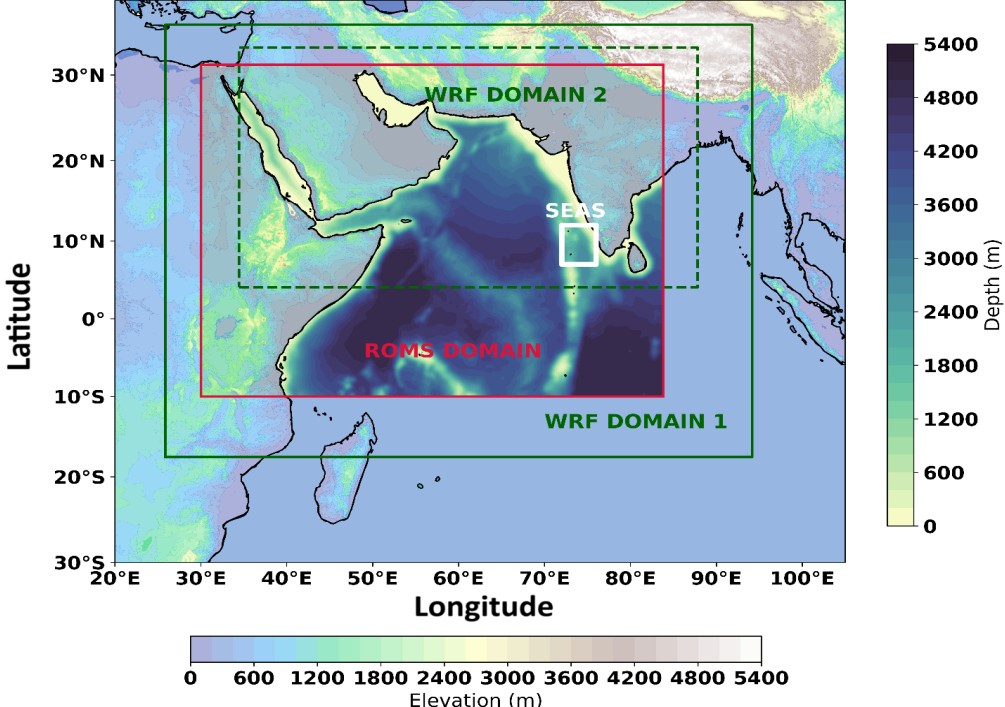

**Figure 1: WRF and ROMS model domains. The WRF domain is the green box, while the red box shows the ROMS model domain. Two domains of WRF are shown in solid and dashed green lines. The white box is the southeastern Arabian Sea (core area of the MWP). Land elevation and ocean Depth are shown in two different contours.**

The WRF domain has an outer domain resolution of 60 km (18ºS to 38ºN & 25ºE to 95ºE) and a nested domain with a 1:3 ratio and consists of 40 levels in the vertical direction. WRF is initialized with ERA5 data. The outer domain's lateral boundary condition is taken from ERA5 at 3-hour intervals. The ROMS model covers the Arabian Sea (10ºS - 30ºN & 30ºE - 85ºE) with a horizontal resolution of $\frac{1^o}{6}$ and 40 sigma vertical levels, 20 of which are in the top 200 m. The lateral boundary in the ROMS model is closed in the north and west and kept open in the south and east. The preconditioning of oceanic conditions before

April is carried into the model through initial and boundary conditions derived from SODA v3.4.2 reanalysis data.

The coupled model is configured in the north Indian Ocean domain, as shown in Fig 1. From the previous studies and our own experience (based on our several background experiments), we observe that the coupled atmosphere-ocean model (part of the COAWST model) would be more accurate when the simulation is up to seasonal. Therefore, to assess the MWP using this coupled model, we simulated the model for three different independent years (2013, 2016, and 2018), which had

distinct MWP characteristics (based on the interannual variation of the MWP area. See Fig. S1 in the supplementary). Each run is initialized separately from April 1 to June 20 each year, and the output is saved daily.

The outputs we used for analysis in this study are from each year's last 51 days of model runs. We leave the remaining outputs for spin-up purposes. The variables are exchanged between WRF and ROMS every 15 minutes. Details of the variable





exchange between the models are found in Prakash & Pant (2017). We named this set of runs the control experiment (CNTRL).
Further, to investigate the factors contributing to the evolution of MWP, we have designed and performed four idealized numerical experiments using the coupled atmosphere-ocean model with one-way coupling mode. In 2018, the MWP area was closer to climatology (Fig. S1 in supplementary). Hence, this year is used as the base year, and in all four sensitivity experiments, the forcings of the two other years are fed into the 2018 control run. The details of the idealized experiments are shown in Table 1.

145                                  Table 1: List of model sensitivity experiments

| Experiment Name | Details |
| --- | --- |
| $S_{ocean2013}$ | The 2013 ocean initial condition replaces the initial condition of the 2018 control experiment. The atmospheric forcing is kept as of 2018. |
| $S_{atmos2013}$ | The 2013 atmospheric forcing replaces the atmospheric forcing of the 2018 control experiment. The oceanic initial and boundary conditions are kept the same as those of 2018. |
| $S_{ocean2016}$ | The 2016 ocean initial condition replaces the initial condition of the 2018 control experiment. The atmospheric forcing is kept as of 2018. |
| $S_{atmos2016}$ | The 2016 atmospheric forcing replaces the atmospheric forcing of the 2018 control experiment. The oceanic initial and boundary conditions are kept the same as those of 2018. |

## 2.3    Mixed Layer Heat Budget

The mixed layer heat budget provides a detailed analysis of factors that can contribute to the change in the mixed layer
temperature and is calculated using the formula outlined in Akhil et al. (2023); Foltz & McPhaden (2009); Girishkumar et al. (2017); Nyadjro et al. (2012); Prakash & Pant (2017); Vialard et al. (2008) and given as:

$$\frac{\partial T}{\partial t} = \frac{Q_{net}}{\rho C_p h} - \left(u\frac{\partial T}{\partial x} + v\frac{\partial T}{\partial y}\right) + H\left[W_h + \frac{\partial h}{\partial T}\right]\left[\frac{T_h - T}{h}\right] + Residuals \qquad (1)$$

Different terms in the above equation's right-hand side (RHS) contribute to temperature tendency differently; the first term on the RHS is the contribution from net heat flux; the second is from horizontal advection; and the third is from vertical process
and entrainment. T is the average temperature of the mixed layer, t is the time (in days), and h is the mixed layer depth (MLD). In this study, the depth at which the subsurface temperature decreases by 1°C as compared to the surface is used as the isothermal layer depth (ILD) (Kara et al., 2000; Shee et al., 2019; Sprintall & Tomczak, 1992). The MLD is calculated following Shee et al. (2019) as follows.



$\quad \sigma_{t(z=h)} = \sigma_{t(z=0)} + \Delta T \frac{d\sigma_t}{dt}$

Where $\Delta T$ is the 1°C temperature criteria for ILD, and $\frac{d\sigma_t}{dt}$ is the coefficient of thermal expansion. Meridional and zonal velocity at MLD is given by u, and v. $Q_{net}$ is the net heat flux. Here, we have not considered the penetrative shortwave radiation below the MLD. H is the Heaviside step function and is expressed as $H = 0$, $if \left[ W_h + \frac{\partial h}{\partial T} \right] < 0$ or $H = 1$, $if \left[ W_h + \frac{\partial h}{\partial T} \right] > 1$. $W_h$ is the vertical velocity. $T_h$ is the temperature just below (5 m) the depth of the MLD. The residual term represents

contributions from other processes, such as diffusion. The units of all the terms here are represented in the °C.day⁻¹.

## 2.4    Potential Turbulent Kinetic Energy (P_TKE):

The production of turbulent kinetic energy (P_TKE) is used to understand the convective mixing caused by the relative contributions of wind stress (wind-forced momentum flux), freshwater flux (E - P) (haline buoyancy flux), and net surface heat flux (thermal buoyancy flux)(Shankar et al., 2016). P_TKE is calculated using the expression given in Han et al. (2001) and

Rao et al. (2002) and expressed as:

$$P_{TKE} = \rho u_*^3 + \left[ -\frac{\alpha 0.5 g k MLD Q_{net}}{C_p} + 0.5 \rho g k MLD \beta (E-P) S_0 \right]$$

Where, $u_* = \sqrt{\frac{\tau}{\rho}}$ is the frictional velocity, $\tau$ is the wind stress, $\rho$ is the density of the seawater (1026 kg.m⁻³), $k = 0.42$ is the Von Kärmän constant, and g is the acceleration due to gravity. E is the evaporation rate, and P is the precipitation. $S_0$ is the salinity of the ML. $Q_{net}$ is the net surface heat flux. The thermal expansion coefficient ($\alpha$) and haline contraction coefficient

($\beta$) are taken as $\alpha$ = -0.00025 K⁻¹ and $\beta$ = 0.00785 psu⁻¹, respectively, following studies by Shankar et al. (2016) and He et al. (2020). $C_p$ (= 4187 J kg⁻¹) is the ocean's specific heat capacity. The first term on the right-hand side of the above equation represents wind stirring. The first term in the bracket is the P_TKE due to thermal buoyancy representing the effect of net surface heat flux, and the second term is P_TKE due to the haline buoyancy, which dictates the impact of freshwater change. A negative thermal buoyancy flux indicates that heat is lost from the ocean, which causes MLD to deepen. On the other hand, the positive

haline buoyancy flux causes more evaporation, which leads to an increase in MLD due to increased salinity (He et al., 2020; Shankar et al., 2016; Shee et al., 2019).

## 3    Results

### 3.1    Model Validation:

The model simulated SST, sea surface salinity, and surface current are subsequently compared with Advanced Very High-

Resolution Radiometer (AVHRR) SST, European Space Agency (ESA) salinity, and Oscar surface current observation data. The spatial resolution of AVHRR, SST, and ESA salinity is 0.25°×0.25°, while the resolution of Oscar current is 0.33°×0.33°.



All of these data are of daily temporal resolution. The simulated results from the coupled model are interpolated to the observational resolution before comparing the biases. These comparisons are done for each model simulated year 2018, 2013, and 2016, as shown in Fig. 2, 3 & 4. The mean results and bias are presented for each of the three years, from May 1 to June

20. The Indian Ocean is well-known for its seasonally reversing monsoon current. The West India Coastal Current (WICC) transports high salinity water from the northern to southern Arabian Sea in May and June and is the primary driver of inter-basin mass transport (Schott et al., 2009; Schott & McCreary, 2001; Shankar et al., 2002). The model simulated surface currents accurately captured the WICC between May and June for all three years (Fig. 4). Furthermore, the representation of the summer monsoon current in the simulation was commendable. The simulated SST effectively captured the cold SST along the Somalia

coast across all the examined years, firmly aligning with AVHRR SST data (Fig 2). The SST bias remained within 1°C in all three experiments except in the northern Arabian Sea, where a cold bias patch appeared in the model simulated SST. This cold bias is attributed to the dry anomalous wind originating from the northwestern region of the South Asian landmass, a pattern also detected in the CMIP models (S. Sandeep & Ajayamohan, 2014). Despite this, the SST bias remained consistent across the study period, with a minimal bias in the SEAS region (black box in Fig. 2). The model simulated surface salinity revealed

a pronounced salinity tongue in the northern Arabian Sea, a feature similarly observed in the ESA salinity data (Fig. 3). The coupled model effectively reproduced the low salinity water in the southeastern Arabian Sea, with the salinity bias primarily within 0.5 psu across the entire domain, except for a few isolated areas reaching 1 psu.

The vertical temperature and salinity profiles from the coupled model have been validated against buoy measurement at  AD10  located at 10.25°N and 72.25°E (Fig. S2). The temperature and salinity from the coupled model aligned well with

the AD10 temperature and salinity (Fig. S3 to S5 in supplementary). Additionally, subsurface high salinity water was evident in AD10 and model simulations for 2018, 2013, and 2016. To further examine the vertical temperature performance of the model, we conducted a detailed temporal correlation analysis of the simulated temperatures for three distinct control experiments, corresponding to the years 2018, 2013, and 2016. This analysis is performed against observational data obtained from the AD10 buoy location, with calculations made at various depths, specifically at 1, 5, 10, 15, 20, 30, 50, and 100 meters.

The results of this statistical assessment are graphically represented in a Taylor diagram (Fig. 5). Across all three control experiments, the standard deviation of the simulated temperature values generally ranged between 0.3 and 1.5, with a few exceptions at certain depths. Additionally, the correlation coefficient between the model simulations and the buoy observations varied from 0.5 to 0.98, indicating a moderate to strong agreement depending on the depth considered.

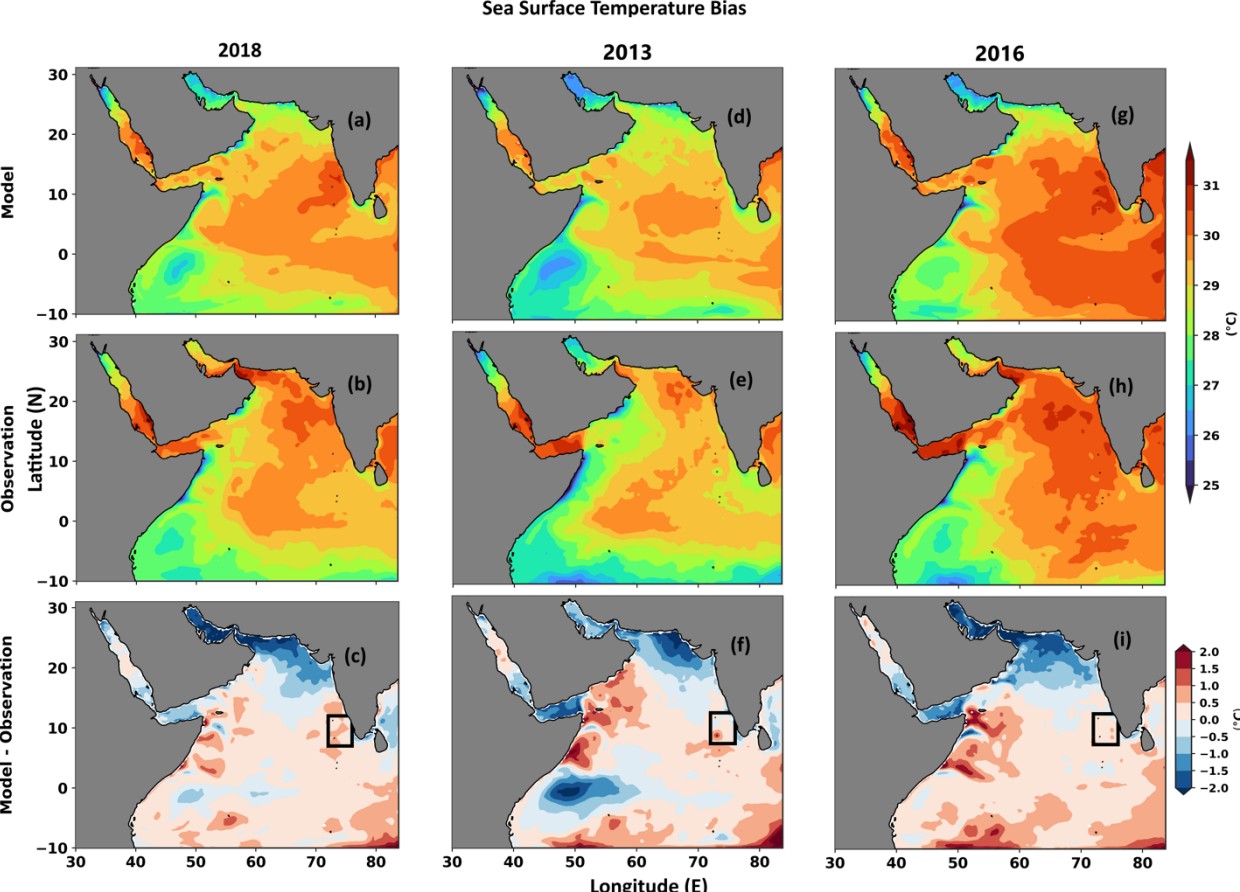


**Figure 2: Comparison of model-simulated sea surface temperature (SST) and NOAA-AVHRR surface temperature. Panels (a), (d), and (g) depict model-simulated SST for the years 2018, 2013, and 2016, respectively. Panels (b), (e), and (h) display NOAA-AVHRR SST for the corresponding years. Panels (c), (f), and (i) illustrate the difference between model output and observation. The black box delineates the domain of interest overlaid on panels (c), (f), and (i).**





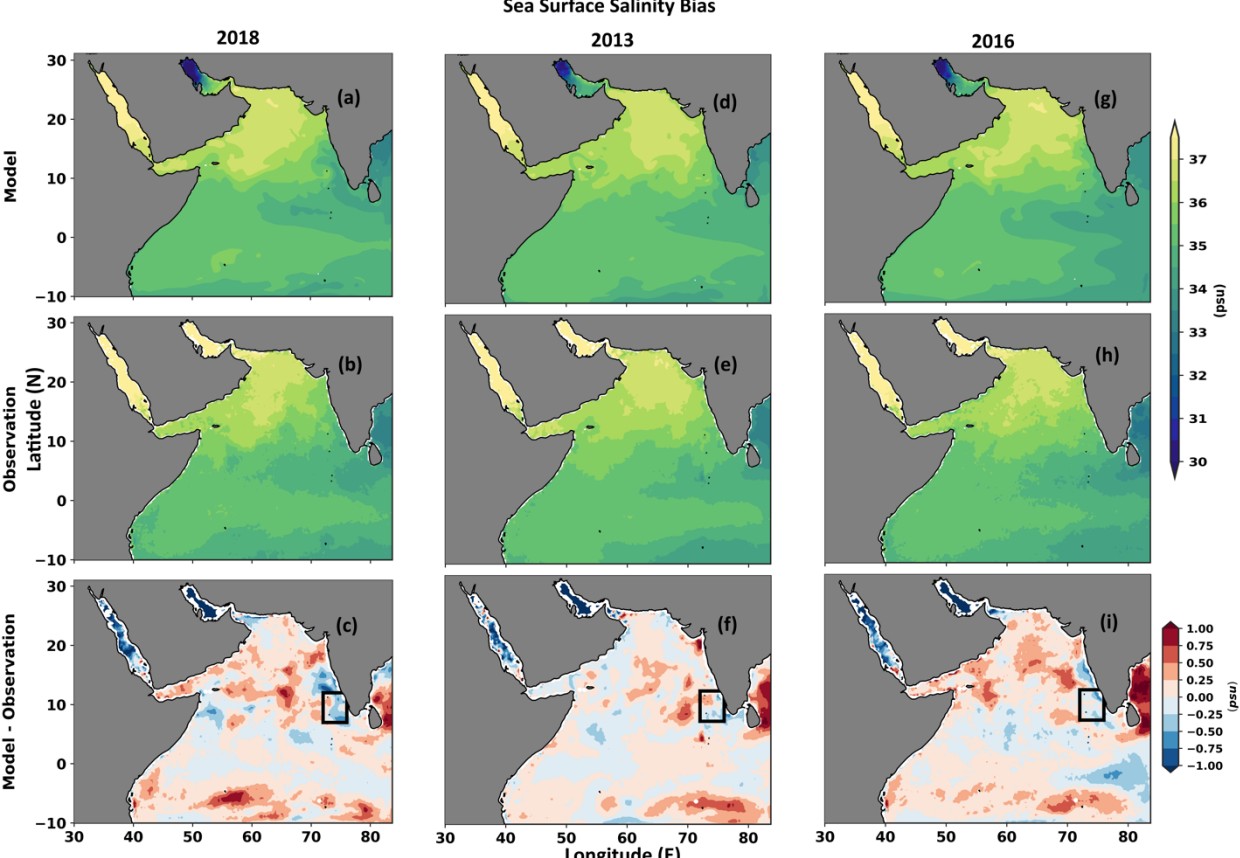


**Figure 3: Comparison of model-simulated sea surface salinity (SSS) and ESA observed sea surface salinity. Panels (a), (d), and (g) depict model simulated SSS for the years 2018, 2013, and 2016, respectively. Panels (b), (e), and (h) display observed SSS for the corresponding years. Panels (c), (f), and (i) illustrate the difference between model output and observation. The black box delineates the domain of interest overlaid on panels (c), (f), and (i).**

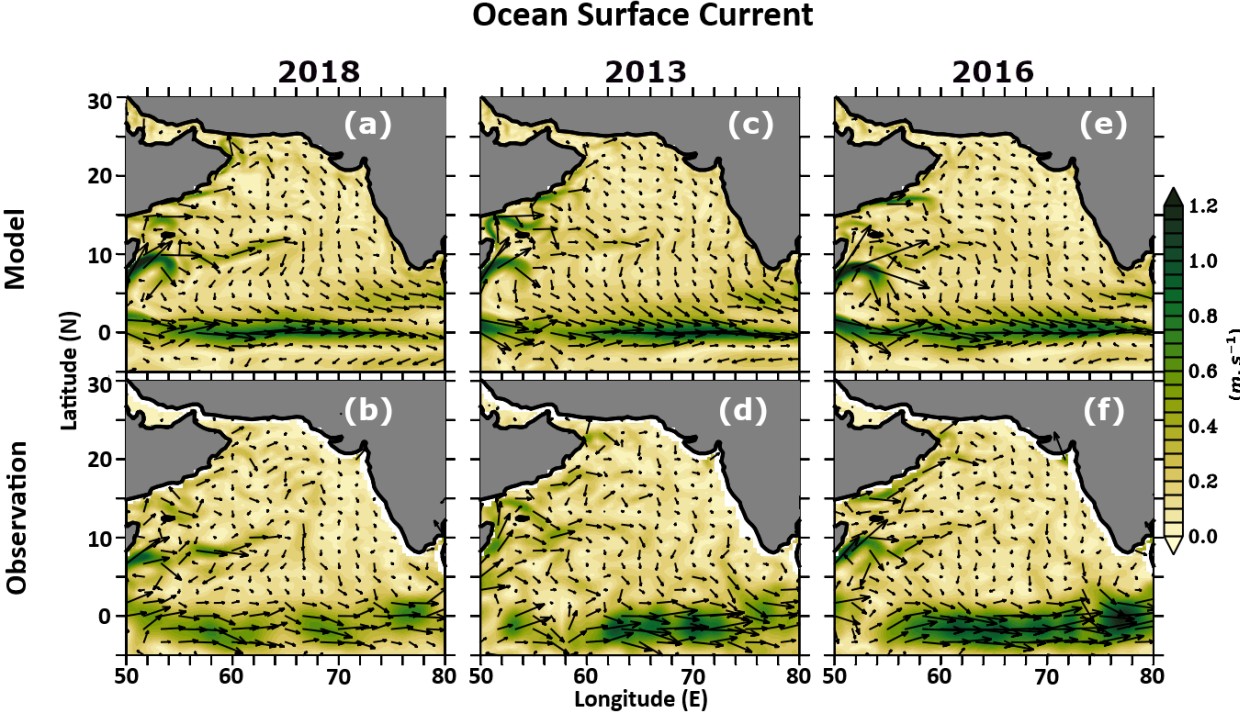

**Figure 4: Comparisons of model simulated currents against Oscar surface current. The first column from the left (a and b) is for 2018, the second is for 2013 (c and d), and the last is for 2016 (e and f). As, the MWP forms near the western coast of India, the current patters are shown for the Arabian Sea, not for the whole model domain.**





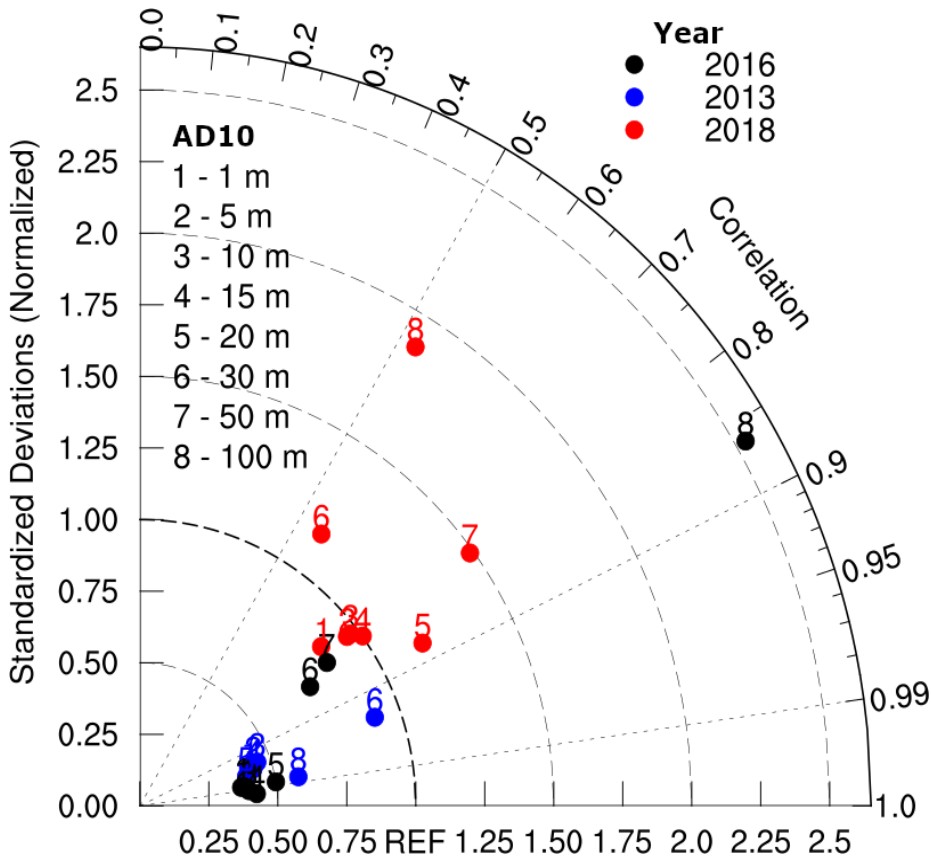

**Figure 5: Taylor diagram showing the temporal correlation and standard deviations of model simulated temperature profile from the three different years of model experiments with respect to measurements at the buoy AD10, the points at 50 m depth are out of the range of axes for the year 2013.**

## 3.2    Ocean surface characteristics during various phases of Arabian Sea Mini Warm Pool

Figure 6 shows the model simulated SST, wind stress, and salinity during the mature and dissipation phases of MWP. The mature phase of the MWP is characterized by the day when the sea surface temperature (SST) within the MWP core (shown by the white box in Fig. 1) reaches its highest magnitude in May. The dissipation day is determined when the MWP SST equalizes that of the surrounding water following its mature phase. Previous studies (Li et al., 2023; Rao & Sivakumar, 1999;
etc.) have used 30°C to identify the MWP. However, a consistent bias of 0.5°C is noticed in SEAS; hence, our study used a threshold of 30.5°C to detect the MWP. Furthermore, in 2018, the SST within the MWP core reached above 31°C on May 20, after which it gradually decreased (Fig. 6a and 6d). During the mature phase, the wind stress over the SEAS remained less. Following May 20, there was a noticeable rise in wind stress, and by June 8, the MWP SST matched the temperature of the surrounding sea. Despite the absence of the MWP in 2013, the same MWP maturity and dissipation day as in 2018 was used for demonstrative purposes (as shown in Fig. 6b and 6e). In 2016, the spatial extent of the MWP was more substantial than in




the other two years (Fig. 6c & 6f). Furthermore, the MWP was prolonged and matured between May 4 and 7, after which it began to diminish. During the mature phase, the wind stress was minimal in the southern Arabian Sea (Fig. 6c). Once the wind stress escalated, the MWP entirely dissipated by June 8.

Salinity in the vicinity of the MWP was lower during its mature phase (<34 psu except in 2013) but increased during its dissipation phase (>35.2 psu). In the MWP core region, salinity was notably less in 2018 compared to the other two years (Fig. 6g to 6i). Kumar et al. (2009) reported that low salinity water is one of the reasons behind the MWP development. Yet, in 2016, while the MWP was in its mature stage, low salinity water (less than 34 psu) was detected south of SEAS, slightly outside the core MWP area (Fig. 6i). Additionally, the north Arabian Sea harbors high salinity water (Kumar & Prasad, 1999), which is transported equatorward along the west coast of India through the WICC and reaches the southern tip of India, where it intrudes into the Bay of Bengal (Vinayachandran et al., 2013). The signal of the southeastward propagation of this high salinity water was evident in the sea surface salinity during the mature to dissipation phase of the MWP in all three years (Fig. 6g to 6l).

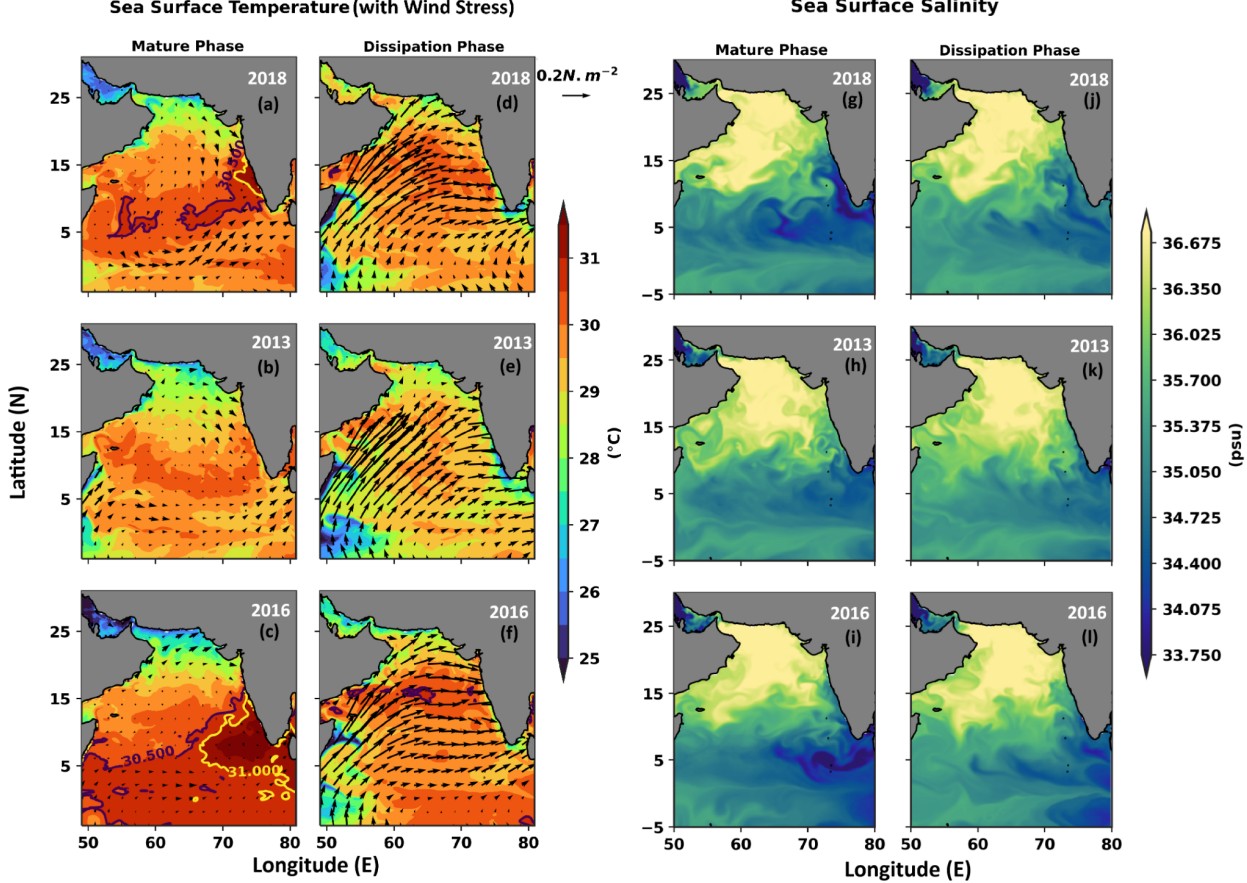

**Figure 6: A comparison of sea surface temperature overlaid by wind stress and salinity during the mature and dissipation phase of the MWP. The mature phase for the mini warm pool is considered on May 20, 2018, 2013, and**





**May 4, 2016. The dissipation phase for the mini warm pool is taken on June 8, 2018, 2013, and 2016. The black and yellow contours in figures (a) to (f) represent the contours of 30.5ºC and 31ºC, respectively.**

In the mature phase of the MWP, there was a positive net heat flux across the Arabian Sea except for its southern flank 265 (Fig. 7a, 7b, and 7c). However, as wind stress intensified during the dissipation phase, the net heat flux in the Arabian Sea transitioned to negative values. Subsequently, four elements of the net heat flux are examined to understand its impact on the regional growth of the MWP in SEAS (Fig. S6 in supplementary). The shortwave radiation flux is always positive and contributes majorly to the net heat flux. Because of the clear sky, the shortwave radiation is higher in April – May (Li et al., 2023). Longwave radiation flux was negative in the Arabian Sea (Fig. S6 in supplementary). The exchange of energy between 270 the atmosphere and ocean is facilitated by turbulent processes, such as sensible and latent heat flux (Large & Pond, 1981). However, these components of the net heat flux could not justify the progress of the MWP in the SEAS.





**Figure 7: Same as Fig. 6 but for net heat flux during the mature and dissipation day of the MWP in 2018, 2013, and 2016.**



Figure 8a to 8c illustrate the vertical temperature of the MWP core area (white box in Fig 1). In 2018 and 2016, the high-temperature water associated with the MWP extended to the MLD during its mature phase (Fig. 8a and 8c). Furthermore, from May 10, 2018, a break in the vertical extent of this high-temperature water (SST>30.5°C) is noticed (Fig. 8a). The vertical extent of the MWP started to shallow from May 8 and regained its depth on May 20. In contrast, the presence of the MWP

water (water with temperature > 30.5°C) is more prolonged in 2016 without any break (Fig 8c). The mixed layer heat budget in the core region of the MWP (72-76°E and 7-13°N) is analyzed to enhance a better understanding of the factors contributing to its expansion and dissipation.

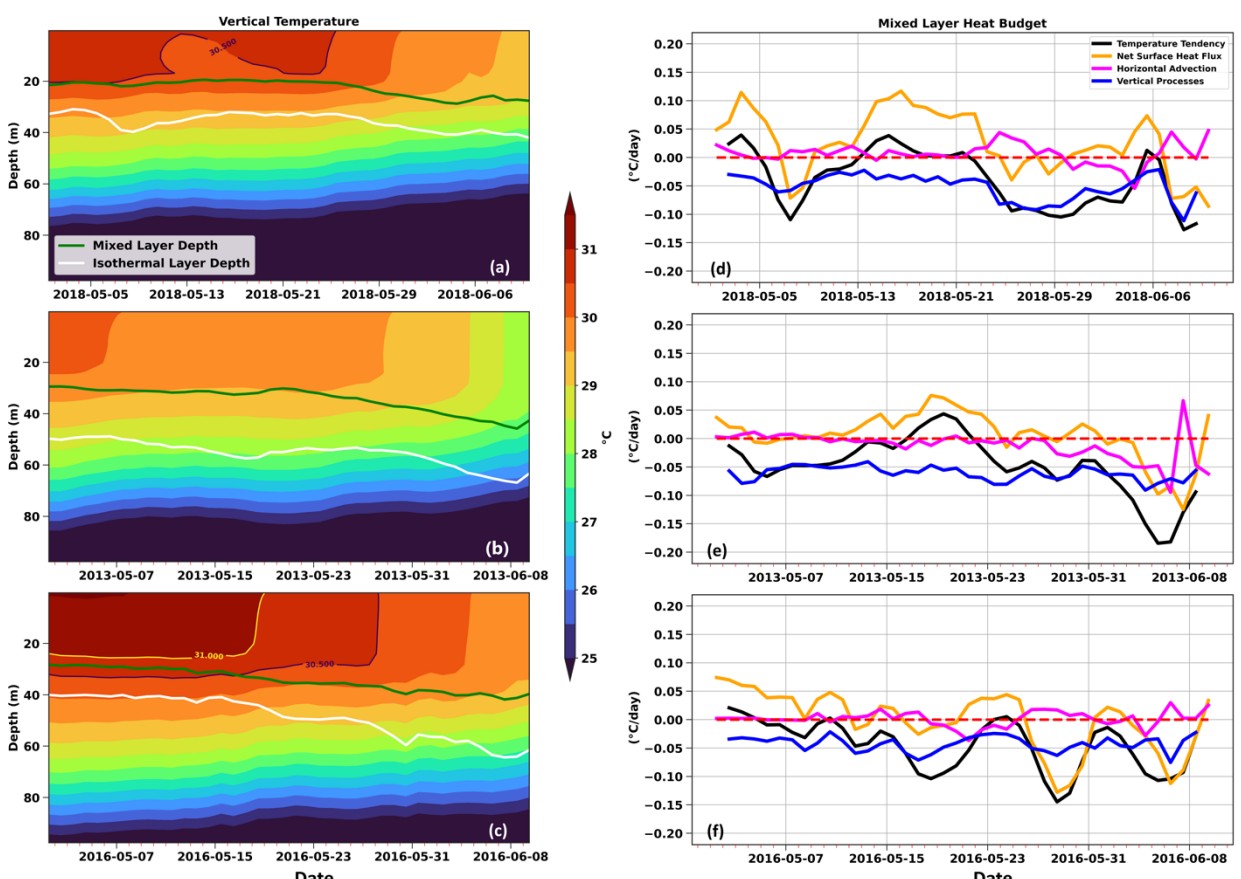

**Figure 8: Panel (a) to (c) shows area-averaged (72- 76ºE and 7-13ºN, the white box shown in Fig. 1) vertical profile of temperature and (d) to (f) shows different terms corresponding to the mixed layer heat budget in the same box. Panels (a) and (d) are for 2018, panels (b) and (e) are for 2013, and panels (c) and (f) are for 2016. The red dashed line suggests the zero line. The black and yellow contours in figures (a) to (c) represent the contours of 30.5ºC and 31ºC, respectively.**



During the mature stage of MWP, the net heat flux positively aided the mixed layer temperature tendency (Fig. 8d to 8f). Li et al. 2023; Mathew et al. 2018; also inferred a similar conclusion. In 2018, the net heat flux supplied to mixed layer temperature tendency by 0.1°C per day (Fig. 8d). Nonetheless, a dip in the net heat flux from 0.1°C.day$^{-1}$ to -0.07°C.day$^{-1}$ was noticed on May 8. This drop in the temperature is noticeable in the vertical temperature as well (Fig. 8a). Even if the MWP was not apparent in 2013, the net heat flux caused a gain of 0.05°C temperature within the mixed layer on May 20 (Fig. 8e). Once it reached the mature phase, the vertical processes had a detrimental impact on the mixed layer temperature during all three years, leading to the dissipation of the MWP. For instance, since May 23, 2018, SEAS witnessed a remarkable decline in mixed layer temperature, attributed to vertical processes (contributes to a maximum of -0.09°C.day$^{-1}$ on May 26), as shown in Fig. 8d. Additionally, the horizontal advection had minimal impact on the development and the dissipation of the MWP in all three years.

### 3.3    Causative Factors

#### 3.3.1    The Role of the Atmosphere and Ocean in the Formation of MWP

This section examines the relative contributions of the ocean and atmosphere to the development of the MWP through four sensitivity experiments (see Table 1 for details). In the 2013 control experiment, the MWP didn't form; however, the $S_{ocean2013}$ experiment revealed the presence of a weak MWP near the coast in the SEAS (Fig. 9). In contrast, the 2016 control experiment experienced anomalously warm conditions, with the MWP SST exceeding 31°C at its core (Fig. 6c). Similarly, the MWP core SST reached a maximum temperature exceeding 31°C in the $S_{ocean2016}$ experiment (Fig. 9). In the MWP core, the initial pre-April ocean temperature was 0.15°C lower in 2013 and 0.35°C higher in 2016 compared to 2018. Following the initial conditions, the MWP core temperature within the MLD was 0.3°C lower in $S_{ocean2013}$ and 0.6°C higher in $S_{ocean2016}$ compared to the 2018 control simulation throughout the mature phase (Fig. S7a and S7c in the supplementary are compared with Fig. 8a). These differences in the temperature solely correspond to the ocean's initial condition before April.

In the $S_{atmos2013}$ experiment, no sign of the MWP was noticed (Fig. 10). Further, in the $S_{atmos2016}$ experiment, the MWP matured from the 1st to the 8th of May (Fig. 10). However, the MWP reached its mature condition in the 2018 control experiment on May 20. Additionally, in the $S_{atmos2016}$ experiment, the MWP became very weak in its area and magnitude on May 20, indicating the influence of the prevailing atmospheric conditions on modulating the MWP's spatial variability.



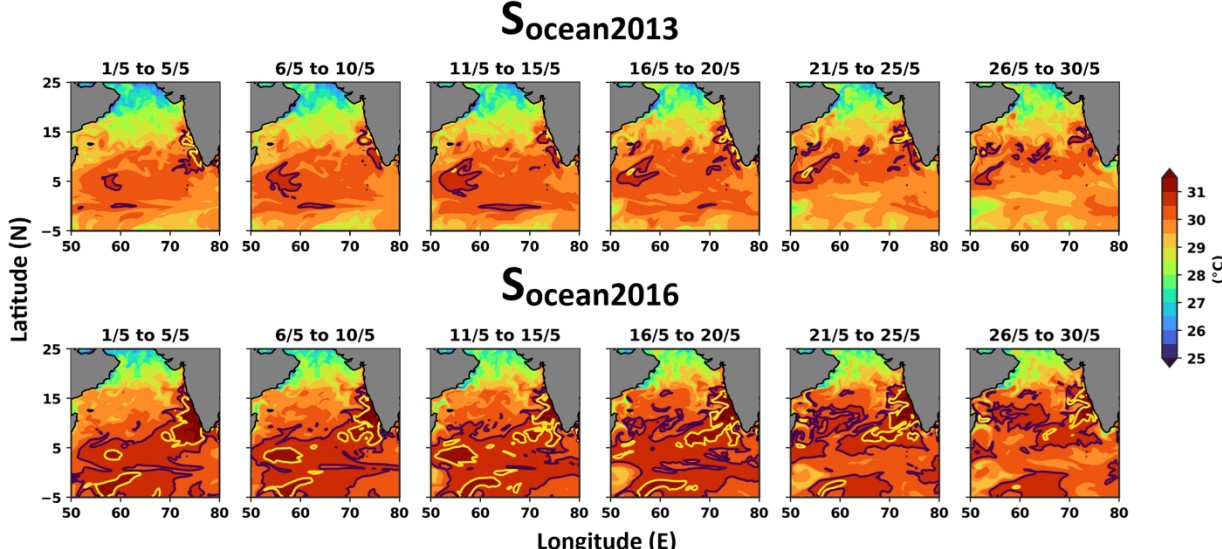

**Figure 9: 5-day average Evolution of SST from May 1 to May 30 for the experiment Socean2013 and Socean2016. In these two experiments, the ocean's initial condition was changed to 2013 and 2016 in the 2018 control experiment. See Table 1 for further details. The black and yellow contours represent 30.5ºC and 31ºC, respectively. The days associated with mean evolution are displayed at the top of each subplot.**

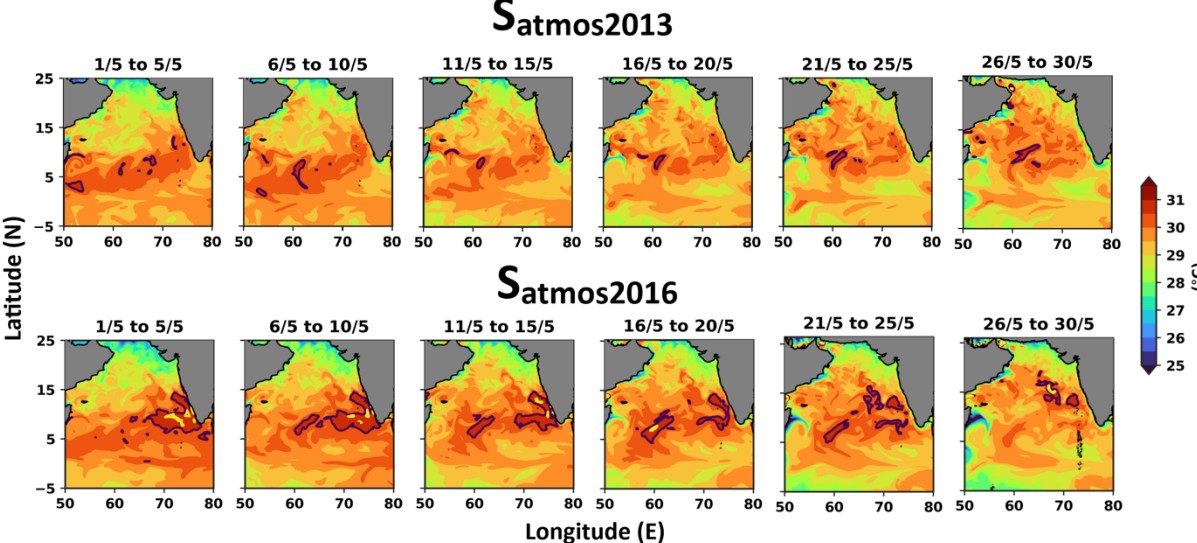

**Figure 10: 5-day average Evolution of SST from May 1 to June 17 for the experiment Satmos2013 and Satmos2016. In these two experiments, the atmospheric forcings were changed to 2013 and 2016 in the 2018 control experiment. See Table 1 for further details. The black and yellow contours represent 30.5ºC and 31ºC, respectively. The days associated with mean evolution are displayed at the top of each subplot.**



Furthermore, the mixed-layer heat budget calculated using the outputs of the sensitivity experiments was compared with the control experiments to investigate the influence of the atmosphere and ocean on the variations in mixed-layer

processes. In experiments $S_{atmos2016}$ and $S_{atmos2013}$, the net heat flux in the mixed layer heat budget followed the net heat flux of 2016 and 2013 in the control experiments (see Fig. 11b and 11d and compare with Fig. 8e and 8f, respectively). These changes in net heat flux resulted from variations in the atmospheric state. The influence of horizontal advection on temperature tendency was negligible across all experiments. Moreover, the atmosphere was the primary driver of the vertical processes within the mixed layer that lead to the dissipation of the MWP. Consequently, once the ocean's initial condition was altered in $S_{ocean2013}$

and $S_{ocean2016}$, a minute change in this vertical process was observed (Fig. 11a and 11c compared with Fig. 8d). However, the vertical processes in the $S_{atmos2013}$ and $S_{atmos2016}$ experiments change substantially as the atmospheric conditions were shifted to 2013 and 2016, and they closely resembled the pattern of the vertical processes in the respective control experiments (see Fig. 11b and 11d and compare with Fig. 8e and 8f correspondingly).

## Mixed Layer Heat Budget



**Figure 11: Area averaged (72- 76ºE and 7-13ºN, i.e., the white box shown in Fig. 1) different components of mixed layer heat budget for four sensitivity experiments, i.e., (a) Socean2013, (b) Satmos2013, (c)Socean2016, and (d) Satmos2016 respectively. As the sensitivity experiments are conducted on the 2018 control experiment, the time reference in the x-axis is similar to that of the 2018 control experiment.**

To better assess the atmosphere's and ocean's relative contribution to the MWP intensity across different experiments, we introduced an index called the MWP intensity index. This index is defined as:



$$MWP\ intensity\ index\ =\ MWP\ area\ on\ mature\ day\ X\ \Delta T_{avg}$$

Where $\Delta T_{avg}$ represented the average temperature change within the MWP area from its initial condition to the mature day.

In the 2016 control experiment, the MWP intensity index reached its highest value, corresponding to the warmest

MWP SST and the most extensive area (Fig. 12). In contrast, the opposite trend was observed in the 2013 control experiment. In the $S_{ocean2013}$ experiment, where the ocean's initial condition was modified from 2018 to 2013, the MWP Intensity Index decreased by 8.5% compared to the 2018 control experiment. However, a significant 82% reduction in the MWP intensity index was found when the atmospheric forcings were altered to 2013 in the $S_{atmos2013}$ experiment, highlighting the adverse effect of the atmospheric environment on the formation of MWP.

Conversely, the $S_{ocean2016}$ experiment, which replaced the 2018 oceanic initial condition with that of 2016, showed a 136% rise in the MWP intensity index compared to the 2018 control experiment if the ocean's initial condition from 2016 replaced that of the 2018 control experiment ($S_{ocean2016}$). However, when the atmospheric condition was adjusted to 2016 in the $S_{atmos2016}$ experiment, the MWP intensity index decreased by 41%, indicating that the ocean precondition was the primary factor in the genesis of the MWP in 2016 and that the atmospheric condition later favored its development (Fig. 12).





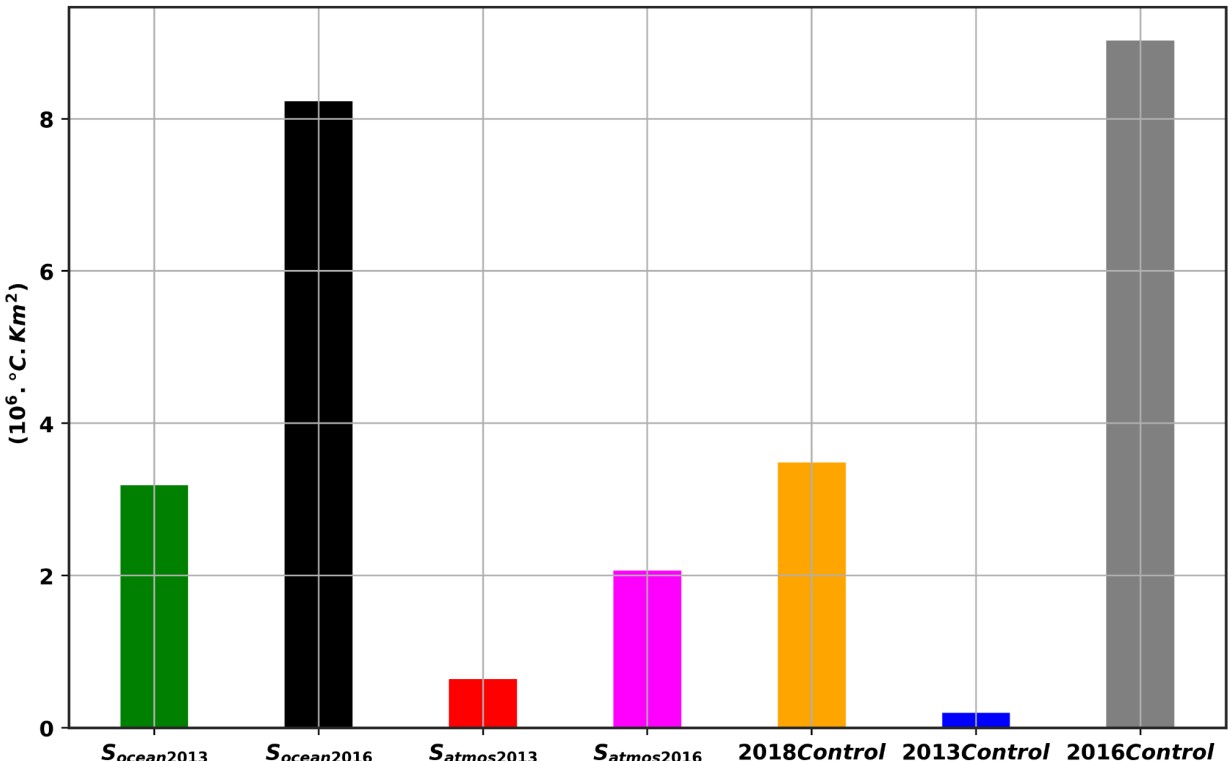

**Figure 12: A comparison of the MWP intensity index in all three experiments during the respective mature phase days in different years. In S$_{ocean2013}$, S$_{ocean2016}$, S$_{atmos2013}$, 2013 control experiment, and 2018 control experiment, the mature day was May 20. In the S$_{atmos2016}$ and 2016 control experiments, the mature day was May 4.**

*3.3.2  **Impact of Ocean Surface Flux  in  the Formation of the MWP***

Once the ocean precondition was met, the atmospheric factors, including net heat flux, wind, and freshwater flux, played a leading role in shaping the spatial extent of MWP. The relative importance of net thermal flux, freshwater flux, and wind on mixing was examined here using potential turbulent kinetic energy (P$_{TKE}$). Contrary to wind stirring, the P$_{TKE}$ by haline and thermal buoyancy flux was minimal in all the experiments, indicating that the wind was the driver for the mixing

in SEAS in May (Fig. 13 to 15). The atmospheric conditions were similar in the 2018 control, S$_{ocean2013}$, and S$_{ocean2016}$ experiments. From 1$^{st}$ to 5$^{th}$ May, the average P$_{TKE}$ caused by wind was more than 8 W.m$^{-2}$; however, in the SEAS, a small shadow zone was formed (Fig. 13). Subsequently, the mixing reduced, and expansion of MWP SST was observed within this shadow zone in 2018 control, S$_{ocean2013}$, and S$_{ocean2016}$ experiments. A similar but much-widened shadow zone was developed in the SEAS from 17$^{th}$ to 20$^{th}$ May, resulting in the largest MWP expansion in 2018 control, S$_{ocean2013}$, and S$_{ocean2016}$ experiments.

Once the southwesterly wind strengthened, the P$_{TKE}$ increased, and the ocean lost heat at the surface (Fig. 13). In the 2013 control and S$_{atmos2013}$ experiments, atmospheric conditions were identical, and the ocean gained heat in the SEAS. Nonetheless, the wind-induced P$_{TKE}$ was more in SEAS, and no such wind shadow zone was developed (Fig 14). Subsequently, the MWP





was absent in these two experiments, indicating that the net heat flux alone can't forge an MWP-like condition. In the 2016 control and $S_{atmos2016}$ experiments, the wind shadow zone was unfurled over a comparatively large area from the 1st to the 8th
of May (Fig. 15). Besides, the ocean received heat during this time in SEAS, making a favorable condition for the development of the MWP once the ocean precondition was favorable (Fig. 15).

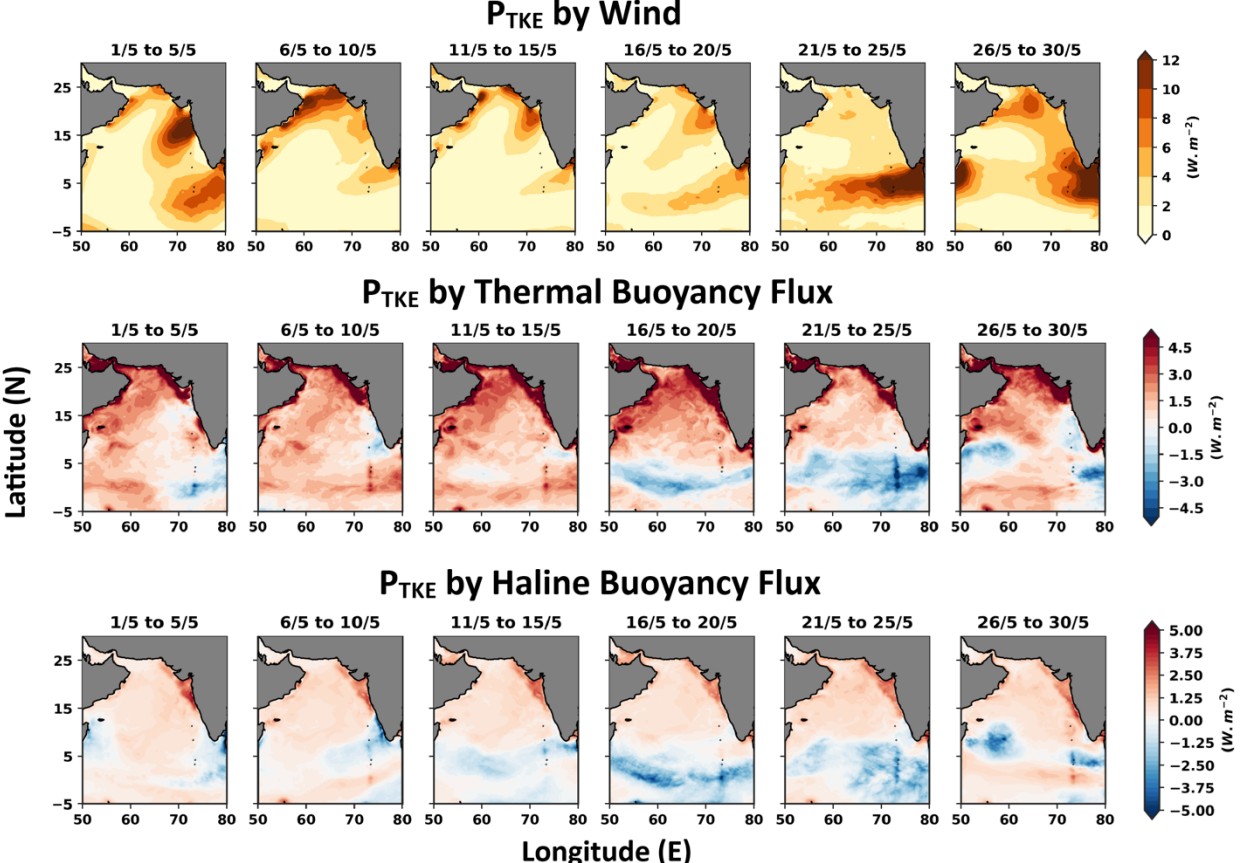

**Figure 13: 5-day average evolution of different components of the production of total kinetic energy in the 2018 control experiment. In S$_{ocean2013}$ and S$_{ocean2016}$ experiments, the atmospheric condition is similar to the 2018 control simulation.**
**Subsequently, this figure also shows the potential turbulent kinetic energy for the experiments S$_{ocean2013}$ and S$_{ocean2016}$. Details of the experiments are depicted in Table 1. The days associated with mean evolution are displayed at the top of each subplot.**






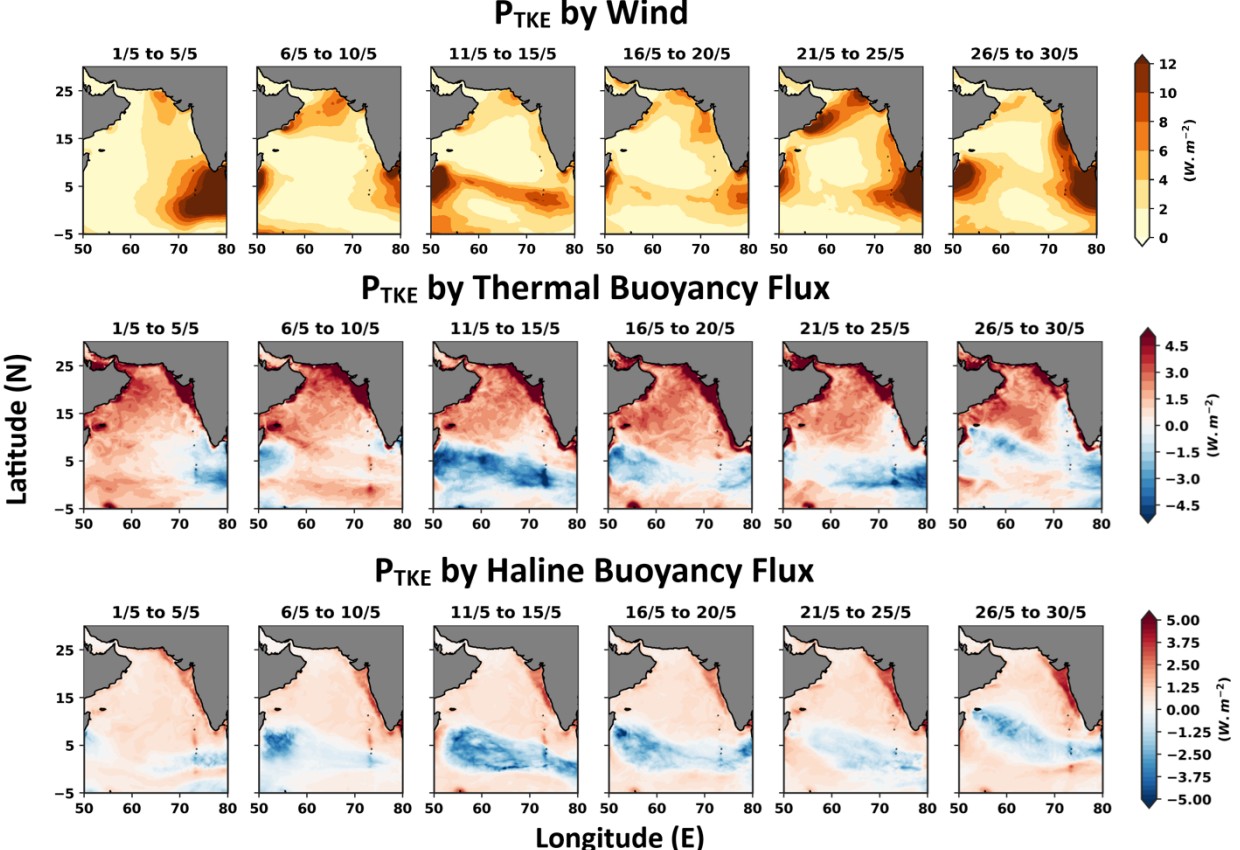

**Figure 14: 5-day average evolution of different components of the production of total kinetic energy in the 2013 control experiment. In S$_{atmos2013}$, the atmospheric condition was similar to that of the 2013 control simulation. Subsequently, this figure also shows the potential turbulent kinetic energy for the experiments S$_{atmos2013}$. Details of the experiments are illustrated in Table 1. The days associated with mean evolution are displayed at the top of each subplot.**




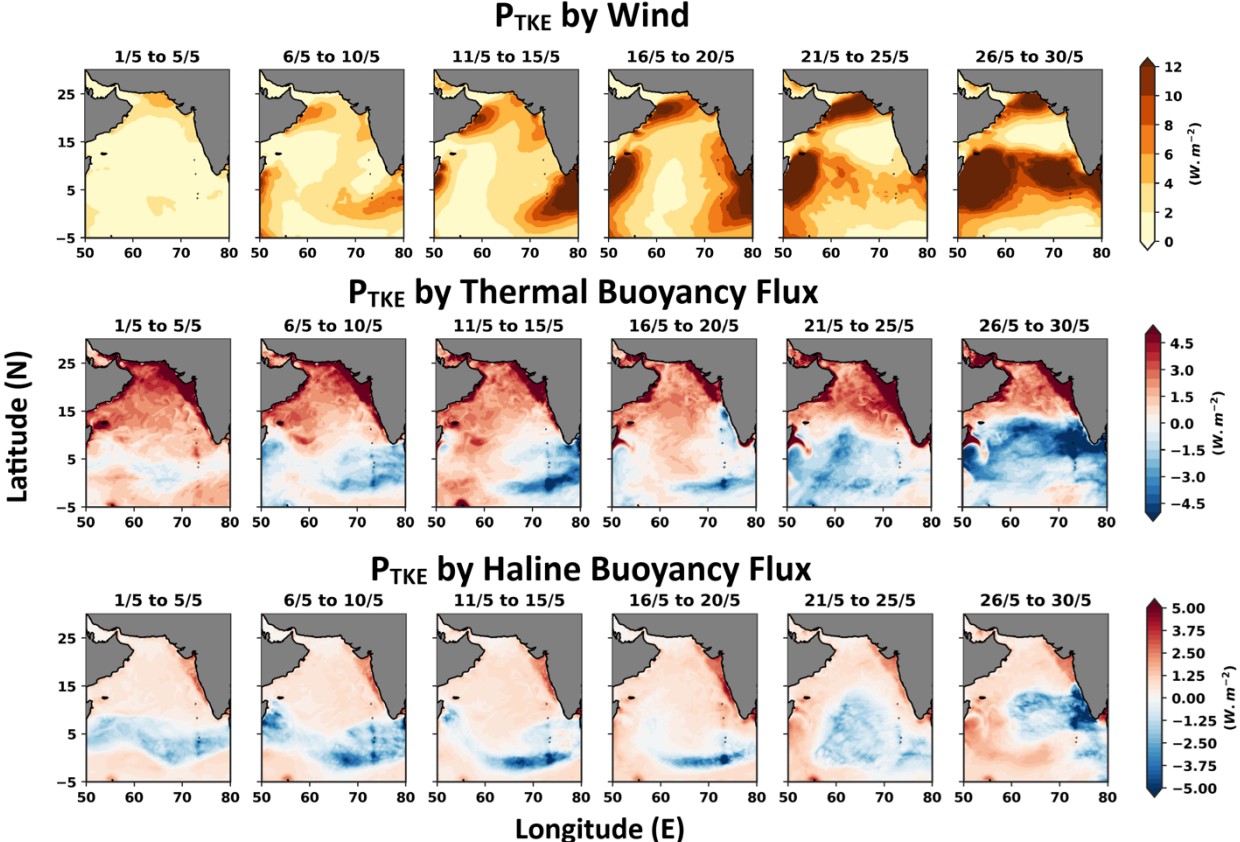

**Figure 15: 5-day average evolution of different components of the production of total kinetic energy in the 2016 control experiment. In S$_{atmos2016}$, the atmospheric condition is similar to the 2016 control simulation. Subsequently, this figure also shows the potential turbulent kinetic energy for the experiments S$_{atmos2016}$. Details of the experiments are depicted in Table 1. The days associated with mean evolution are displayed at the top of each subplot.**

## 4   Conclusion

A coupled atmosphere-ocean model has been employed to investigate the formation and evolution of the MWP in the southeastern Arabian Sea. The model is simulated for three independent years: 2013, 2016, and 2018. The simulated results were able to reproduce the MWP features and showed good agreement with observational data. Analysis of the mixed-layer heat budget (Fig. 8) revealed that net heat flux is the primary driver of the MWP development, contributing significantly to the increase in mixed-layer temperature. It contributed more than 0.1°C per day to the increased temperature throughout the MWP development phase. However, net heat flux alone did not fully account for the regional expansion of warm SST observed in the SEAS. Once the MWP reached its mature phase, vertical processes negatively affected temperature tendencies by -0.08°C/day, leading to its rapid dissipation. Additionally, four sensitivity experiments further elucidated the roles of oceanic





and atmospheric conditions in the MWP formation. The changes in ocean initial conditions significantly impacted the
magnitude of the MWP core temperature, resulting in 0.6°C higher in $S_{ocean2016}$ and 0.3°C lower in $S_{ocean2013}$ compared to the
2018 control experiment (Fig. S7a and S7c are compared with Fig. 8a), indicating the influence of the ocean initial condition
in the MWP intensity. The change in atmospheric conditions has altered the roles of net heat flux and vertical processes in
temperature tendencies of the heat budget. This vertical process was responsible for the dissipation of the MWP. Subsequently,
the MWP intensity index revealed that shuffling of the atmospheric and oceanic initial conditions of the strong MWP year to
the year of the MWP whose intensity was identical to climatology (see the details of the experiment $S_{atmos2016}$ and $S_{ocean2016}$)
resulted in a substantial rise of 41% and 136% in MWP intensity index respectively, which demonstrated that in the strong
MWP year, the pre-April ocean condition played a considerable role in the development of the MWP in May which was further
supported by the prevailing atmospheric conditions. This contradicts previous studies, such as Kurian and Vinayachandran
(2007), which suggested that MWP development in May was independent of the pre-April ocean conditions. Furthermore, the
exchange of initial conditions of ocean and atmospheric forcings from the weak MWP years to the year of the MWP whose
intensity is identical to climatology ($S_{atmos2013}$ and $S_{ocean2013}$ experiment) resulted in a decrease of 82% and 8.5% in the MWP's
intensity, respectively, suggesting that in the weak MWP year, although the ocean preconditions were met, it's the atmospheric
forcing that restricted the development of the MWP in SEAS.

Later, $P_{TKE}$ was computed to study the mixing in the SEAS. The influence of wind induced $P_{TKE}$ on mixing far
surpassed that of $P_{TKE}$ resulting from thermal and haline buoyancy flux. Further, a wind shadow zone with less $P_{TKE}$ was
witnessed in SEAS, and the MWP advanced within this zone. However, in experiments such as the 2013 control and $S_{atmos2013}$
experiment, this shadow zone was not present, and the MWP did not develop, although the ocean preconditions were met,
indicating that the wind shadow zone was a key factor in the MWP's advancement within the SEAS.





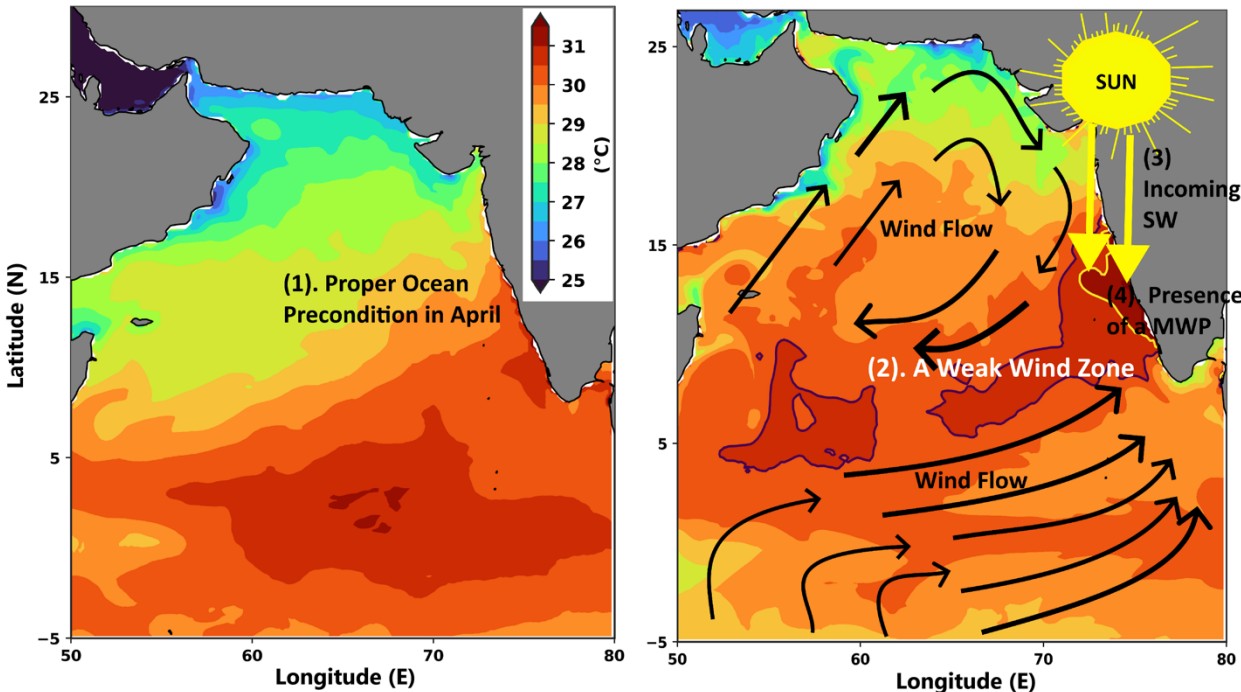

**Figure 16: Schematic of the atmospheric and oceanic conditions associated with the formation of the MWP. The numbering denotes the importance of that process in the formation of MWP. For instance, the ocean precondition (1) is the first requirement for MWP's genesis. Later, a weak wind zone (2) with less mixing traps the incoming shortwave radiation (3) in SEAS, which results in the formation of the MWP (4). Both the subplots have same color bar levels. The black and yellow contours represent 30.5ºC and 31ºC, respectively.**

In conclusion, the formation mechanism of the MWP is a fully coupled process driven by both ocean preconditions and atmospheric forcings (Fig. 16). The coupled atmosphere-ocean numerical model proved to be an essential tool for understanding this complex interaction. However, further modeling efforts are required to better understand the air-sea interaction during the MWP mature phase. Furthermore, 2016 was a strong El-Nino year, and the impact of winter salinity stratification in such intense El-Nino years on the pre-April SST is a subject for future scope of research. Moreover, given that the wind shadow zone emerges during a year with high MWP SST, we hypothesize that this zone and the corresponding increase in MWP SST could be associated with the onset of the Indian Summer Monsoon. However, additional investigation is necessary, which is beyond the scope of the current study.



**Data Availability:**

SODA3 data is used to give initial and boundary condition to the ocean part of the numerical model (Carton et al., 2018), whereas ERA5 data is used for the atmospheric boundary and initial condition (Hersbach et al., 2020). NOAA-AVHRR SST data is downloaded from https://www.ncei.noaa.gov/products/avhrr-pathfinder-sst (Saha et al., 2018). OSCAR current data is used to validate the model (Bonjean & Lagerloef, 2002). All the model output data may be available on request to the author.

**Software Availability:**

Authors gratefully acknowledge USGS for making the COAWST numerical model available openly. In this study, python is used for the graphical plots. In addition, the Gibbs Seawater package (McDougall & Barker 2011) (https://www.teos-10.org/software.htm#1) is utilized to compute basic oceanic parameters. Further, wrf-python is utilized to handle the WRF output (Ladwig 2017). Software/programs related to the study may be available from the corresponding author upon request.

**Author Contributions:**

SPL ran the numerical models, analyzed the results and wrote the first draft of the manuscript. KRP helped in analysis and manuscript editing. VP reviewed the manuscript and supervised the work.

**Competeing Interests:**

The contact author has declared that none of the authors has any competing interests.

**Acknowledgments**

This work is a part of SPL's doctoral thesis work. SPL sincerely acknowledge the financial support of the Prime Minister's Research Fellowship (PMRF). The computational resources acquired through the high-performance computing (HPC) cluster at the Indian Institute of Technology Delhi are acknowledged.

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
