# Peer review of "An evaluation of the Arabian Sea Mini Warm Pool's advancement during its mature phase using a coupled atmosphere-ocean numerical model"

_EGUsphere, 2024_

## Author Comment (AC1)

**Author's Response to Reviewer's Comment**

***Manuscript No. -*** *egusphere-2024-2848*

***Title:*** *"An evaluation of the Arabian Sea Mini Warm Pool's advancement during its mature phase using a coupled atmosphere-ocean numerical model"*

**Comments to Reviewer - 1**

*Prasad Lahiri et al. employs the use of a coupled ocean-atmosphere regional model (ROMS + WRF) to examine processes contributing to the mini warm pool in the Arabian Sea. They examine three events of the warm pool in their model and attempt to determine the relative contributions between ocean and atmospheric processes driving the strength and dissipation of these events. While I find the concept of the study particularly interesting and potentially a good contribution to literature in the future. Nevertheless, significant revisions must be made before this manuscript can be reconsidered for publication.*

*As detailed throughout the comments below, there is a need to restructure the manuscript text in general to make it clear and concise. There is an extreme overuse of conjunctive adverbs such as "moreover", "however", "nonetheless" and many others that is in several cases is erroneous. There are missing references in several parts. The model description and validation are lacking. There are also several statements throughout the results section that do not appear to be supported by the figures. For example, the provided plots qualitatively comparing with a buoy do not support statements that the model compares well with the buoy data. Second, the statement on line 296: Figure 8f shows that the net surface heat flux is the dominant term driving the heat tendency with contributions from vertical processes remaining largely the same throughout the time series. It is my impression that several of the comments below can be addressed with a thorough revision of the text.*

**Reply:**

*We sincerely thank the Reviewer for your tremendous effort and time in reviewing the manuscript. Your insightful feedback has undoubtedly helped to improve the quality and the scientific rigor of the work. In response to your suggestions, we have restructured the manuscript and removed several conjunctive adverbs in the updated manuscript. Also, the interpretation regarding the influence of the net surface heat flux on the mini-warm pool has been corrected. Following your suggestions, we have addressed each of your comments individually, as listed below. The revised*

*manuscript incorporates these changes. Your comments are presented in black, and our responses are provided in blue italic font. For comments containing multiple queries, we have addressed each point as bullet points for clarity.*

1. *Section 2.2: I find the description of the model components lacking and insufficient. For example: no information is given on the time-step of each component, no information is provided on how the open boundary conditions are specified except for the datasets, the horizontal advection scheme used in ROMS, and other important configuration details. Furthermore, the authors jump back and forth describing the model components which is quite confusing. I'm also curious about the selection of nested domains that are not fully overlapping each other between the components (WRF domain 2 does not fully overlap with the ROMS domain). See the references below for examples on complete model descriptions:*

   a. *Olabarrieta, M., Warner, J. C., Armstrong, B., Zambon, J. B., & He, R. (2012). Ocean–atmosphere dynamics during Hurricane Ida and Nor'Ida: An application of the coupled ocean–atmosphere–wave–sediment transport (COAWST) modeling system. Ocean Modelling, 43, 112-137.*

   b. *Castruccio, F. S., Curchitser, E. N., & Kleypas, J. A. (2013). A model for quantifying oceanic transport and mesoscale variability in the Coral Triangle of the Indonesian/Philippines Archipelago. Journal of Geophysical Research: Oceans, 118(11), 6123-6144.*

   c. *Ross, A. C., Stock, C. A., Adcroft, A., Curchitser, E., Hallberg, R., Harrison, M. J., ... & Simkins, J. (2023). A high-resolution physical-biogeochemical model for marine resource applications in the Northwest Atlantic (MOM6-COBALT-NWA12 v1. 0). Geoscientific Model Development Discussions, 2023, 1-65.*

   d. *Seijo-Ellis, G. G., Giglio, D., Marques, G. M., & Bryan, F. O. (2024). CARIB12: A Regional Community Earth System Model/Modular Ocean Model 6 Configuration of the Caribbean Sea. EGUsphere, 2024, 1-48.*

**Reply:**

♦ *We sincerely thank the Reviewer for this valuable comment. As suggested, we have restructured the 'Model Details' section. Initially, we provided a general overview of the model, followed by a detailed discussion of the specific configurations and schemes applied*

*in our study. Additionally, we have elaborated on the model setup in coupled mode and described the exchange of variables between the atmospheric and oceanic components. These updates can be found in lines 107–116 and 140–154 of the revised manuscript.*

♦ *The WRF atmospheric model domain was configured such that the parent WRF domain is larger than the ROMS model domain, as recommended by Warner et al. (2010). For the inner nested domain, we specifically focused on covering the region of the mini warm pool. This inner domain was chosen to encompass a slightly larger area of the mini warm pool to better resolve its processes at finer resolution and to account for inflow and boundary reflections. Therefore, there is no compromise in addressing the hypothesis, regardless of whether the innermost WRF domain fully overlaps the ROMS domain. The nested domain adequately spans the entire Arabian Sea and the southern part of the Indian landmass, aligning with our primary area of interest. Furthermore, we conducted sensitivity experiments to evaluate the impact of domain size. While a larger nested domain was tested, the results within the mini warm pool showed no significant differences. Considering this, and to maintain computational efficiency, we opted for the current domain configuration. This coupled model setup is both effective and optimal for our study objectives.*

2) *Table 1 and sensitivity experiments. Are the open boundary conditions for the first time step also replaced along with the initial conditions? If not, wouldn't this generate noticeable discrepancies and noise near the boundaries?*

**Reply:**

*We agree with the Reviewer that if the boundary conditions are not replaced, initial noise at the boundary could travel inside the domain. For this reason, we have also changed the boundary conditions along with the initial condition in the sensitivity experiments. However, we missed to add this information in the previous manuscript. This information is updated in Table 1 in the revised manuscript. Thank you for pointing out this issue.*

3) *The authors omit important citations in several places, for example no reference is given to the ERA5 and SODA reanalysis datasets or the ROMS and WRF models. These should be cited in the text, not just the Data availability statement.*

    a. *ERA5: Hersbach, H., and Coauthors, 2020: The ERA5 global reanalysis. Q.J.R. Meteorol. Soc., 146, 1999–2049, https://doi.org/10.1002/qj.3803.*

    b. *SODA: Carton, J.A., G.A. Chepurin, and L. Chen (2018), SODA3: a new ocean climate reanalysis, J. Climate, 31, 6967-6983, https://doi.org/10.1175/JCLI-D-18-0149.1*

*Reply:*

*Thanks for pointing out the important citation that we missed it. Now, we have added these citations in the revised manuscript in the respective places. Please see the lines 138-139 and 144 - 145 in the revised manuscript.*

4) *136-137: It would be much clearer and more concise to say: "The first month of each simulation is used for spin-up and not included in the analysis." On that note, is a 1-month spin-up realistically enough for this case? Are the ocean boundaries nudged or sponge layers used to help with initial noise at the ocean boundaries?*

*Reply:*

*In the revised manuscript, we have included this statement in lines 167-168. We used the reanalysis assimilated product of initial and boundary conditions for both ocean and atmospheric models. Therefore, in these conditions, the model was able to quickly reach a steady state, particularly within the upper few hundred meters (0–200 m). The extent of the mini warm pool is confined to the mixed layer depth, which remains within 50 m during its mature phase across all control and sensitivity experiments (refer to the 30.5°C contour in Fig. R1). Thus, a 1-month spin-up is sufficient in the configured numerical model to simulate the mixed layer processes effectively. Additionally, the ROMS model, coupled with WRF, has been shown to perform well for shorter simulation periods. Extending the spin-up time would necessitate initializing the model earlier than April (as the analysis period is May to June), which would not only increase computational costs but also introduce model biases. For these reasons, we have retained the 1-month spin-up period in our study.*

[Figure]

*Fig R 1 Area averaged (72- 76ºE and 7-13ºN, i.e., the core MWP region) vertical temperature for three control ((a) 2018 control experiment, (b) 2013 control experiment, and (c) 2016 control experiment) and four sensitivity experiments ((d) $S_{ocean2013}$, (e) $S_{ocean2016}$, (f)$S_{atmos2013}$, and (g) $S_{atmos2016}$). In the sensitivity experiments, the oceanic and atmospheric conditions have been changed to various years; thus, only the day and month are kept on the x-axis.*

♦ *We have not used any sponge layers or nudging at the boundaries. We have used radiation boundary conditions at the open boundary for tracers and momentum. This information is added in lines 150-151.*

5) *Line 139: "We named this set of runs the control experiment (CNTRL)." What set of runs? The ones in Prakash & Pant (2017)? If so, the simulations must be described to some extent here, I have no idea what they did there.*

***Reply:***

*Thank you for pointing out this confusing statement. We didn't refer to Prakash & Pant (2017) for the control experiment, but the variables exchanged between WRF and ROMS. Further, we have removed the citation from here for clarity. This statement is restructured in the revised manuscript. Please see lines 163-169 for clarification.*

*6) All datasets used should be described under Section 2. Data and Methodology not in the Results section. That includes the validation datasets and description of any processing done to them for the purposes of the validation comparison.*

**Reply:**

*We sincerely thank the Reviewer for bringing this point to our attention. We have added a new 'Data' sub-section in the 'Data and Methodology' section to discuss the data. Please see lines 85 to 96.*

*7) Lines 190-202: Figure 4 results are described before Figure 2 and 3.*

**Reply:**

*Thank you for pointing out this. We have re-arranged the text here and discussed Fig. 2 first and then Fig. 3 and 4. Please see lines 215-229 in the revised manuscript.*

*8) Figure 4 should have a differences panel like Figures 2 and 3. This is important because there seems to be biases in the magnitude and spatial extent of some features. Current vectors could also be included in the difference panels similar to Figure 4 in Liu et al. (2015).*

> *a Liu, Y., Lee, S. K., Enfield, D. B., Muhling, B. A., Lamkin, J. T., Muller-Karger, F. E., & Roffer, M. A. (2015). Potential impact of climate change on the Intra-Americas Sea: Part-1. A dynamic downscaling of the CMIP5 model projections. Journal of Marine Systems, 148, 56-69.*

**Reply:**

*We thank the Reviewer for this suggestion. We have updated Figure 4 to closely align with the style of Liu et al. (2015). Additionally, per Reviewer 2's comment, we have focused the plot on a zoomed-in Indian West Coast region rather than displaying the entire domain.*

*9) Lines 194-: "The simulated SST effectively captured the cold SST along the Somalia coast across all the examined years, firmly aligning with AVHRR SST data (Fig 2). The SST bias remained within 1°C in all three experiments except in the northern Arabian Sea, where a cold bias patch appeared in the model simulated SST." While the cold tongue is present in the model, the statement as it is currently written is not supported by Figure 2 which shows some of the largest SST biases occur along the Somalian coast (particularly in 2016). Similarly, there are*

*biases above 1degC (positive and negative) in other parts of the domain, not just in the Arabian Sea. While these biases are likely acceptable (the authors must convince the reader they are), it is still important to recognize them properly.*

**Reply:**

*We thank the Reviewer for pointing out this point.*

- ♦ *Numerical models have some biases, and they become more evident in the reanalysis data forced regional numerical models. Our configured model is no exception to that. Patches of warm SST bias in the boundary and near the coast in all the years have been observed. However, most of these biases are within 1.5-2ºC. Besides, the southeastern Arabian Sea is a very complex region due to the inter-basin transport of water and associated tracers from the Arabian Sea to the Bay of Bengal. This region has a very low bias in temperature and salinity, indicating that our configured numerical model adequately captured the inter-basin transport of water and associated tracers. As our focus is the mini warm pool in the southeastern Arabian Sea, this inter-basin transport becomes more important than the patches of warm and cold biases in the Somalia coast and the northern Arabian Sea.*

- ♦ *We also agree with the Reviewer regarding the proper identification of these biases. Hence, the warm SST bias near the Somalia coast has been recognized. Furthermore, a strong cold SST bias in 2016 near the tropical west Indian Ocean is also identified and reported in the revised manuscript. Please look at the lines 215-221 in the revised manuscript.*

10) *Lines 196-198: "This cold bias is attributed to the dry anomalous wind originating from the northwestern region of the South Asian landmass, a pattern also detected in the CMIP models (S. Sandeep & Ajayamohan, 2014)." This doesn't make sense to me. Have you tested this is true in your model? CMIP models are free-running global climate models while you are running a reanalysis forced model. Unless you tested this is true, there is no reason at all to believe the same bias would exist in your forced simulation. Sandeep and Ajayamohan (2014) show that this SST bias results from biases in the representation of large-scale circulation on CMIP models. Those biases would likely not be present in your model and very constrained in the reanalysis forcing.*

**Reply:**

*Thank you for pointing this out. We agree with the Reviewer regarding the difference between the global and reanalysis forced models. Thus, the bias which are in the CMIP6 likely may not be the same in our regional coupled model. Therefore, we have removed this information from the revised manuscript.*

11) *Lines 203-206: The figures in the supplemental materials must have a panel with the differences. The current qualitative comparison does not support the statement that the model results aligned well with the Buoy data. This part of the validation is important as the model must represent the temporal and vertical evolution of properties which are important to the processes the authors aim to understand better.*

 (a) *Figure S3 SST: shows noticeable difference in the time extent and magnitude of the warm temperatures and the shallowing of colder waters towards the end of the simulation.*

 (b) *Figure S3 SSS: shows even more noticeable discrepancies between the model and buoy data. The buoy misses a notable high salinity water mass around June 2018. The near-surface distribution of salinity is also quite different.*

 (c) *Similar comments generally apply to Figures S4 and S5.*

**Reply:**

*We have now shown the difference in the temperature and salinity for all three years in S3 to S5. The buoy data has some missing values, which were interpolated in the last version of the manuscript; however, in the revised version, we haven't incorporated such interpolation technique and kept the actual data. In this revised version, the point closest to the AD10 location is used in the numerical model for the comparison, unlike the previous version, where the temperature and salinity of the numerical model are averaged within a 1 X 1° box created around the AD10 location and then compared. Thus, these comparisons are much more precise. In addition, we have revised the text in the manuscript in lines 230-247 to address this validation in detail. Thank you so much for your comments, which improved our results.*

12) *The authors do not provide a reference to the buoy data.*

**Reply:**

*We have somehow missed including this information, which have been added in the "Data" subsection as well as in the "Data availability" section. Please see lines 93 to 96 and 486 to 487 in the revised manuscript. Thank you for reminding us of this point.*

13) Lines 206-208: Some additional information would be useful. Is this taking from a single grid point in the model closest to the buoy location? Or is this a horizontal average? If it is a horizontal average, over what region?

**Reply:**

*We thank the Reviewer for this suggestion. We compared the model data with the nearest 1 X 1º box to the AD10 location in the numerical model vertical temperature and salinity in the previous version of the manuscript; however, in this revised version, the point closest to the AD10 location is used. We have included this information in the revised manuscript. Please see lines 230-232.*

14) Line 138: Be consistent with the terminology, is it dissipation day or phase. One implies several days the other one a specific day. These terms seem to be used interchangeably throughout the manuscript which is confusing.

**Reply:**

*Thank you for pointing out this error. In the present study, the dissipation day is defined as the day when the SEAS averaged temperature becomes the same as that of its surroundings. Thus, it is a day. On the other hand, the dissipation phase comprises a few days from the mature day to the dissipation day. For this updated information, please see lines 278-281 in the revised manuscript.*

15) The definition for the mature phase in Lines137-138 say "...is characterized by the day when the sea surface temperature (SST) within the MWP core (shown by the white box in Fig. 1) reaches its highest magnitude in May." Why only in May? What if it reaches the highest SST within the MWP on June?

**Reply:**

*Typically, wind conditions remain weak from late April to early May, accompanied by clear skies during this period. From late May to early June, the Indian Summer Monsoon sets in over the subcontinent. The associated southwesterly winds induce strong upwelling along the west coast of India, leading to surface cooling. Consequently, a strong and intense MWP develops in May and*

*dissipates as the southwesterly winds strengthen. For reference, the seasonality of SST in the SEAS during strong MWP year is illustrated in Fig. R2.*

[Figure]

*Fig R 2 Seasonality of SST in SEAS (72 - 77ºE & 7-13ºN) averaged over strong MWP years i.e., 1998, 2003, 2005, 2010 and 2016 .*

*16) Line 241: the use of "Furthermore" is not correct here.*

**Reply:**

*We are sorry for this error. This sentence is restructured in the revised manuscript. Please see lines 283-284 in the subsection' Ocean surface characteristics during various phases of Arabian Sea Mini Warm Pool'. Apart from this, we have also concise our study by removing unnecessary furthermore, nonetheless, and similar conjunctive adverbs in different places. We thank the Reviewer for pointing this out.*

*17) Line 242: "...the wind stress over the SEAS remained less." Less than what? Sentence is incomplete.*

**Reply:**

*In this sentence, we meant that the wind speed over SEAS is comparatively lower than that of the surrounding area. This sentence is reframed in the revised manuscript (see lines 284 – 286 in the updated manuscript).*

*18) A table detailing the mature day and dissipation day for each event would help the reader. The table can include the threshold used to define each.*

***Reply:***

*We have added a table detailing the mature and dissipation days for different years. We thank the Reviewer for this valuable suggestion. Please see Table 2 for this information in the revised manuscript.*

*19) The authors often omit the year when writing dates which makes the reading more confusing that it already is with the back and forth between the different years.*

***Reply:***

*We sincerely apologize for this confusion. The date and the years are being updated in the respective places in subsection 'Ocean surface characteristics during various phases of Arabian Sea Mini Warm Pool' in lines 277 to 315 in the revised manuscript. However, in the sensitivity experiments, as the atmospheric forcings and oceanic conditions correspond to different years, we mentioned the experiment name rather than the year in the updated manuscript's 'Causative Factors' section.*

*20) The domain of interest should be included in ALL figures and panels showing maps like shown in panels c, f and g of Figure 2.*

***Reply:***

*Thanks for the suggestion. We have included the domain of interest in all the figures in the updated manuscript.*

*21) Figure 6. Since the description of this figure in the text largely relies on the difference in the conditions between the mature and dissipation phases the authors could consider showing a similar figure with the difference between the time periods (i.e. dissipation – mature). This would be much more informative and relatable to the text in most of Section 3.2.*

***Reply:***

*We agree with the Reviewer that a difference panel will make the interpretation easier for readers. Hence, a panel containing the difference between the dissipation and mature day is included in*

*Fig. 6, 7, and 8 in the revised main manuscript and Fig. S6, S7, S8, and S9 in the revised supplementary. Thank you for this valuable advice.*

22) Line 255-257: "The signal of the southeastward propagation…" Without the surface currents I can't tell the low salinity signal is propagating southwestward, could as well be that the source of that low salinity is weaker and thus its extent is less.

***Reply:***

*We concur with the Reviewer regarding the variability of the source of the high-salinity water. Kumar & Prasad (1999) reported that this high salinity water formed in the northern Arabian Sea during winter due to convective mixing. Thoppil et al. (2022) studied and concluded a substantial interannual variability of the source of the high salinity water in the northern Arabian Sea during contrasting monsoon seasons. However, in the aforementioned statement in the manuscript, we meant that this high salinity water is transported southeastward from the mature to dissipation day in each year (Fig. R3). Thus, we are not comparing between the years but between the mature and dissipation days within each year. However, this southeastward transport of the high-salinity water is not the main objective of our study. Hence, to reduce the redundancy, we have removed this statement from the revised manuscript.*

**Ocean Current**

*Fig R 3 Anomaly of the current speed along with direction between the dissipation and mature day in (a) 2018, (b) 2013, and (c) 2016. The black box is the MWP core region.*

23) *Line 271: "However, these components of the net heat flux could not justify the progress of the MWP in the SEAS." How do the authors reach this conclusion? it seems obvious to me that the latent heat flux is an important contributor to the progression of the MWP: the pattern of the net surface heat flux during the dissipation phase is very similar to the combined patterns of latent and surface heat fluxes. This is not surprising, as wind stress increases it drives evaporation at the surface which acts to cool the ocean surface via release of latent heat. In fact, Figure 8f shows that the heat tendency is largely explained by the net surface heat fluxes.*

***Reply:***

*We strongly agree with the Reviewer on this point. We have misinterpreted the influence of the net surface heat flux on the MWP formation. We have corrected this misinterpretation in the updated manuscript. See lines 358-373 under the subsection 'The Role of the Atmosphere and Ocean in the Formation of MWP' and 438 to 445 and 471 to 478 in the 'Conclusion' section for the corrected interpretation. We apologize for the misinterpretation.*

24) Line 292-294: "In 2018, the net heat flux supplied…" This needs re-writing as it implies this is true for the full time series in Figure 8d which is not.

**Reply:**

*We thank the Reviewer for the thorough inspection. We have corrected this statement in the revised manuscript (see the line 361-363).*

25) Lines 296-300: While vertical mixing has a negative contribution to the heat tendency (i.e. cooling effect) that is true for the full time series. The vertical mixing curve remains largely flat with no noticeable trend driving the large variations shown for the heat tendency. The statement the authors make is only somewhat true for 2018. In 2013 its clear that contributions from horizontal advection and net surface fluxes drive most of the heat tendency during the dissipation phase. In 2016, the net surface heat fluxes also drive the heat tendency during the dissipation phase with the contribution from vertical mixing largely uniform throughout the time series.

**Reply:**

*We appreciate the Reviewer's insightful comment regarding the cooling influence of the net surface heat flux. As suggested, we agree that after the MWP matures, increased wind speed leads to enhanced evaporation (latent heat loss) and strong mixing. Additionally, the moisture-rich southwesterly wind reaches the SEAS (after a few days of the MWP mature day) and causes cloudiness, which blocks the incoming shortwave radiation. These factors collectively result in a negative impact of net surface heat flux on the MWP. In response, we have revised the manuscript to include a detailed analysis of the mixed layer heat budget, highlighting this negative influence (see lines 358-373, particularly lines 368–373). Furthermore, we have incorporated this information into the conclusion section (see lines 438-445 and 471-478) for a comprehensive discussion.*

*26) The panels in Figures 9 and 10 need numbering.*

***Reply:***

*We have numbered the panels in Fig. 9 and 10. Thank you for this advice.*

*27) Line 307: "...the intial pre-April ocean temperature…" Is the author referring to the initial condition mean SST? Please clarify this in the text.*

***Reply:***

*The referenced temperature represents the average temperature within the SEAS region (denoted by the white box in Figs. 9 and 10) up to the mixed layer depth (MLD) in the ocean's initial conditions. This temperature is subsequently compared with sensitivity experiments to assess the impact of ocean preconditions. For example, in the 2016 (2013) ocean initial condition, the MLD-averaged SEAS temperature was 0.35°C higher (0.15°C lower) than in 2018. Similarly, when the ocean's initial and boundary conditions were altered to 2016 (2013) in 2018, the MWP became 0.6°C warmer (0.3°C cooler). To enhance clarity, we have revised the relevant statement in lines 329-333 of the updated manuscript. We kindly request the Reviewer to review the changes.*

*28) Lines 304-315: The description along these lines is extremely confusing and difficult to follow within the context of what the authors aim to describe. I would encourage rewriting. Start by reminding the reader what the first sensitivity experiment was, describe the results of the experiment, then connect with the results of the control simulation and what the results from the sensitivity experiment mean. Then do the same for the second sensitivity experiment.*

***Reply:***

*We sincerely thank the Reviewer for this insightful suggestion. In line with the Reviewer's advice, we have simplified the paragraph for better clarity. The revised version can be found in lines 325-342 of the manuscript.*

*29) Line 314-315 seem to contradict the statements made before about net heat fluxes and vertical mixing in the control experiment, which I already commented about in #24 above.*

***Reply:***

*We thank the Reviewer for this comment. Following the suggestion of the Reviewer, we have re-written this whole section, and, in the process, this contradictory information is removed from the revised manuscript.*

30) *Figure 11: each panel should identify to which experiment is corresponds to. I shouldn't need to read the caption to find this information.*

*Reply:*

*We have included the information of the experiment in each panel. We have updated this not only in Fig. 11 but also in Fig. S10 in the supplementary. Thank you so much for your suggestion.*

31) *It would be more useful and informative if Figures 11 and 8 were combined.*

*Reply:*

*We have merged Figures 11 and 8, as well as Figures 8a–8c and Figure S7 from the supplementary materials. The latter has been relocated to the supplementary section, as Figure S10 in the revised version. The merging of Figures 11 and 8 (now Figure 11 in the revised manuscript) allowed us to rewrite this section, combining the results of the control and sensitivity experiments. We appreciate the Reviewer's suggestion, as it has enhanced the fluency and coherence of the manuscript. Please refer to lines 358–373 for the updated information*

32) *Line 333-334: "Moreover, the atmosphere was the primary driver of the vertical processes within the mixed layer that lead to the dissipation of the MWP." I'm not sure what the author means by this. How is the atmosphere the primary driver of vertical mixing processes in the ocean? Furthermore, the net surface heat fluxes are the main contributor to the tendency 11d, with combined contributions from all terms in Figure 11b. Furthermore, as detailed before, the contribution by vertical processes remains mostly the same throughout the time series, so its hard to understand in Fig.11 b and d how vertical processes are the drive the dissipation phase.*

*Reply:*

*We can understand the Reviewer's concern regarding this statement. We meant here that the vertical processes continuously have a detrimental effect on the temperature tendency of the MWP and could facilitate its dissipation. However, the Reviewer fairly pointed out that the net surface heat flux has the dominant influence on the MWP temperature tendency, which we also agree with. Once the wind speed in the southeastern Arabian Sea intensifies, it causes latent heat loss from the surface along with a decrease of incoming shortwave radiation due to the cloudy sky associated with the arrival of the moisture-rich southwesterly wind. This results in a reduction of the net*

*surface heat flux. Besides, the wind along the southwest coast of India is favorable for upwelling, and once the wind speed increases, so does the upwelling (Rao et al., 2008; Shah et al., 2015). Thus, the atmosphere, especially wind, drives SEAS's vertical processes. Nevertheless, the net surface heat flux is the primary driver behind the variation of the temperature tendency from the mature to the dissipation day in the southeastern Arabian Sea. We have rewritten lines 368-373 in the revised manuscript to add this information. Thank you for your in-depth review.*

*33) Lines 336-338 seem to me repetitions of Lines 330=331?*

***Reply:***

*We thank the Reviewer for the in-depth inspection. In the revised manuscript, we have rewritten the whole section (line 368-372 in the revised manuscript) to add comments 31 and 32. As a result, these lines have been removed.*

*34) Figure 12-15. Experiment must be identified in the figure in addition to the caption.*

***Reply:***

*We thank the Reviewer for this insightful suggestion. In the SEAS, the production of turbulent kinetic energy ($P_{TKE}$) driven by wind has a significantly greater impact compared to $P_{TKE}$ generated by thermal and buoyancy fluxes (Fig. 13). Accordingly, we have retained only the wind induced $P_{TKE}$ in the revised main manuscript (Fig. 14) while the $P_{TKE}$ caused by haline and thermal buoyancy fluxes has been moved to the supplementary material (Figs. S11 and S12). Experiment details are now included in all these figures, and panels have been appropriately numbered. Additionally, the MWP core region is marked with a black box in these figures for clarity.*

*35) Lines 373-374: shadow seem like an odd way to describe an area of high Ptke. Keep it simple and call it what it is.*

***Reply:***

*We thank the Reviewer for this query. The MWP expanded in the southeastern Arabian Sea in a region where the $P_{TKE}$ induced by wind is substantially low, which we referred to as the "wind shadow zone." It is important to note that we did not label the region with high $P_{TKE}$ as the shadow zone. Please refer to lines 408-419 in the revised manuscript for the updated information.*

*36) Figures 13-15 are all discussed in a single paragraph. I would recommend combining (by eliminating unnecessary panels). The authors do not describe and/or comment on the panels associated with the thermal buoyancy flux or haline buyoyancy flux driven turbulent energy.*

*Reply:*

*Thank you for pointing this out. The $P_{TKE}$ caused by haline, or thermal buoyancy flux, has less magnitude than the $P_{TKE}$ caused by wind (Fig. 13 in the revised manuscript). Due to this reason, we have focused on the $P_{TKE}$ caused by wind. Thus, following the Reviewer's advice, the $P_{TKE}$ caused by the wind for all the experiments is combined and kept in the revised main manuscript (Fig. 14), and the $P_{TKE}$ caused by haline, and thermal buoyancy flux is moved to the supplementary (Fig. S11 and S12 in supplementary).*

*37) Line 409: "...revealed that net heat flux is the primary driver of the MWP development.." this statement (which I agree with) is not consistent with statements made in the results section (see comment 24 and others). Specifically line 271.*

*Reply:*

*We thank the Reviewer for pointing this out. The interpretation related to the influence of the net surface heat flux on the MWP temperature tendency was not appropriate in the previous version of the manuscript. We have updated this information in the revised manuscript in the respective places. We request the Reviewer to please have a look at lines 358-373 under the subsection 'The Role of the Atmosphere and Ocean in the Formation of MWP' and 438 to 445 and 471 to 478 in the 'Conclusion' section for the corrected interpretation. We apologize for the misinterpretation.*

*38) Lines 411-413: "However, net heat flux alone did not fully account…" You mixed layer budget (Figure 8) does not support the statement that after the mature phase, vertical mixing in the ocean leads to a rapid dissipation of the MWP, except for 2018 to some extent.*

*Reply:*

*We apologize for the misinterpretation. The answer to this comment is similar to the previous one. We have misinterpreted the influence of the net surface heat flux and the vertical processes on the MWP development in the earlier version of the manuscript. However, we have updated this information in the revised manuscript in the respective places. We request the Reviewer to please have a look at lines 358-373 under the subsection 'The Role of the Atmosphere and Ocean in the*

*Formation of MWP' and 438 to 445 and 471 to 478 in the 'Conclusion' section for the corrected interpretation. We apologize for the misinterpretation.*

39) *"Net heat flux" should always be "net surface heat flux".*

*Reply:*

*We thank the Reviewer for pointing this out. We have replaced the Net heat flux with the net surface heat flux in the respective places.*

40) *Line 414: "...significantly impacted..." what defines a significant impact in this case? There is no statistical analysis done to determine significance. Furthermore, one resulted in an increase and the other in a decrease. This is rather inconclusive.*

*Reply:*

- ♦ *We sincerely thank the Reviewer for this suggestion. As we have not done any statistical test here, we have replaced the 'significantly impacted' by 'substantially impacted' in the revised manuscript (see lines 435-438) in the 'Conclusion' section.*

- ♦ *In the $S_{ocean2013}$ and $S_{ocean2016}$ experiments, the ocean initial and boundary conditions in the 2018 control experiment are replaced by the 2013 and 2016 ocean initial and boundary conditions, respectively. In 2013, the MWP was weak, and by replacing the ocean initial and boundary condition in the 2018 control experiment with the 2013 ($S_{ocean2013}$ sensitivity experiment), we observed a weaker MWP than in the 2018 control experiment. Similarly, in 2016, the MWP was intense, and by substituting the ocean initial and boundary condition in the 2018 control experiment to 2016 ($S_{ocean2016}$ sensitivity experiment), we observed a stronger MWP than in the 2018 control experiment. The strength of the MWP changes following the ocean precondition, and thus, we have concluded that the ocean precondition plays a dominant role in the formation of the MWP. We hope this clears the confusion regarding the influence of the ocean's initial condition on the MWP formation.*

41) *Lines 423-424: "This contradicts previous studies, such as Kurian and Vinayachandran (2007), which suggested that MWP development in May was independent of the pre-April ocean conditions." This needs further elaboration and comments by the authors. What are the differences between this study and the Kurian 2007 study that may lead to these different results?*

*Reply:*

*We thank the Reviewer for this advice. Following the comment from the other Reviewer, we have removed this statement from the main manuscript.*

42) *Figure 16 is potentially a nice way to wrap-up the manuscript. But the author make no effort to summarize and describe the formation and dissipation mechanisms of the MWP as shown in the figure. Describe in 1-2 simple sentences the processes as shown in Figure 16.*

*Reply:*

*Following the Reviewer's suggestion, we have provided a detailed description of the MWP genesis as illustrated in the schematic. This explanation can be found in lines 471-476 of the revised manuscript. We sincerely thank the Reviewer for this insightful suggestion.*

*References:*

*Kumar, S. P., & Prasad, T. G. (1999). Formation and spreading of Arabian Sea high-salinity water mass. Journal of Geophysical Research: Oceans, 104(C1). https://doi.org/10.1029/1998jc900022*

*Liu, Y., Lee, S. K., Enfield, D. B., Muhling, B. A., Lamkin, J. T., Muller-Karger, F. E., & Roffer, M. A. (2015). Potential impact of climate change on the Intra-Americas Sea: Part-1. A dynamic downscaling of the CMIP5 model projections. Journal of Marine Systems, 148. https://doi.org/10.1016/j.jmarsys.2015.01.007*

*Prakash, K. R., & Pant, V. (2017). Upper oceanic response to tropical cyclone Phailin in the Bay of Bengal using a coupled atmosphere-ocean model. Ocean Dynamics, 67(1). https://doi.org/10.1007/s10236-016-1020-5*

*Rao, A. D., Joshi, M., & Ravichandran, M. (2008). Oceanic upwelling and downwelling processes in waters off the west coast of India. Ocean Dynamics, 58(3–4). https://doi.org/10.1007/s10236-008-0147-4*

*Shah, P., Sajeev, R., & Gopika, N. (2015). Study of upwelling along the west coast of India-A climatological approach. Journal of Coastal Research, 31(5). https://doi.org/10.2112/JCOASTRES-D-13-00094.1*

Thoppil, P. G., Wallcraft, A. J., & Jensen, T. G. (2022). Winter Convective Mixing in the Northern Arabian Sea during Contrasting Monsoons. *Journal of Physical Oceanography, 52(3).* https://doi.org/10.1175/JPO-D-21-0144.1

Warner, J. C., Armstrong, B., He, R., & Zambon, J. B. (2010). Development of a Coupled Ocean-Atmosphere-Wave-Sediment Transport (COAWST) Modeling System. *Ocean Modelling, 35(3).* https://doi.org/10.1016/j.ocemod.2010.07.010

---

## Author Comment (AC2)

**Author's Response to Reviewer's Comment**

**Manuscript No. -** egusphere-2024-2848

**Title: "**An evaluation of the Arabian Sea Mini Warm Pool's advancement during its mature phase using a coupled atmosphere-ocean numerical model"

**Comments to Reviewer - 2**

Review of "An evaluation of the Arabian Sea Warm Pool's advancement during its mature phased using a coupled atmosphere-ocean numerical model" by S. P. Lahiri, K. R. Prakash, and V. Pant, submitted to EGUsphere.

This paper aims to investigate the temporal progression of the Warm Pool using observations and a regional coupled model (COAWST but without waves). The simulations are quite short (~ 3 months) and are targeted at just one season. WRF component is run at 20km (inner nest) and ROMS is 1/6deg. The experimental and analysis approaches include control and sensitivity experiments for different years, mixed layer temperature budgets, and potential TKE analysis.
The paper has many interesting points and addresses an important climate topic. The experiments are useful, including the various helpful sensitivity experiments. However, there are some major limitations for which I give a Major Revision.

**Reply*:***

*We sincerely thank the Reviewer for your time and effort in thoroughly reviewing the manuscript. Your insightful comments have significantly contributed to enhancing the quality of the manuscript. Following your suggestions, we have restructured the manuscript accordingly. Your comments have been addressed individually, as outlined below, and the corresponding revisions are incorporated into the manuscript. Your comments are presented in black, and our responses are provided in blue italic font. For comments containing multiple queries, we have addressed each point as bullet points for clarity.*

**Major points**

1. The simulations are rather short (April 1st to June 20th). My interpretation of lines 72-74 and 131-133 is that the coupled model develops biases when simulations are longer than a season (please clarify this).

It is possible than by April 1st, the mature phase has already started to develop, as a response to surface fluxes. In this case, the ocean initial conditions on April 1st are a function of the prior surface fluxes of the warming period. Your findings that ocean initial conditions on April 1st are important may not be inconsistent with Kurian and Vinayachandran (2007). If you had initialized on February 1st or March 1st, your statement on lines 423-424 ("contradicts… Kurian and Vinayachandran (2007)") would be reasonable, but for your current set up of April 1st initialization I think this line referring to Kurian and Vinayachandran (2007) should be removed.

Further, you analyze the heat budget from early May to June, but the warm pool has already considerably developed by early May. E.g. in your Fig. 8d,e,f there are only very brief periods in May where the temperature tendency is positive, it is mostly negative, a weakening. I realize you want to avoid the spin-up period, but perhaps you could do a heat budget anyway for April 1st to June and see if you capture any development of the warm pool (i.e. longer period of positive temperature tendency).

**Reply:**

♦ *In this study, we used a regional ocean-atmosphere coupled model where the WRF model was the atmospheric component. The WRF model is generally used for short-term simulation (ranging from a few days to seasonal scales). Based on previous studies and our own experiments, extending WRF simulations over longer periods tends to increase the biases, which can significantly affect the ocean model outputs. Also, our study was focused on investigating the relative contributions of the atmosphere and ocean to the development of the MWP throughout the mature phase. Due to this, we ran the coupled model from April 1st to June 20th in 2013, 2016, and 2018, respectively. Extending the simulation period beyond this was not feasible due to two primary factors: (1) the high computational cost and (2) the progressive increase in model biases.*

♦ *We appreciate the Reviewer's comment. The Reviewer brought up an intriguing topic on the origins of the Arabian Sea Mini Warm Pool (MWP). We agree with the Reviewer that*

*the process of forming the MWP may begin well before April. However, there is disagreement within the scientific community over the exact timing of the mechanisms that contributed to the establishment of the MWP. Akhil et al. (2023); Kurian & Vinayachandran (2007); Mathew et al. (2018) suggested that the development of the MWP does not depend on the antecedent winter stratification and thus, ocean pre-condition has very little influence on the MWP genesis, whereas Durand et al. (2004); Gopalakrishna et al. (2005); Hareesh Kumar et al. (2009); Masson et al. (2005); Nyadjro et al. (2012) reported that the stratification does have a prominent influence on the MWP formation and without the winter stratification the MWP strength could decrease by $0.5^oC$ (Masson et al., 2005). Thus, the present study aims to decipher the impact of the ocean and atmosphere on the MWP during its mature phase, as well as the elements that contribute to the MWP's dissipation. We have concluded that atmospheric processes, particularly wind patterns, determine the spatial variability of the MWP. Nonetheless, the ocean pre-conditions before April have a major effect on MWP strength.*

♦ *We agree with the Reviewer that net surface heat fluxes during pre-monsoon have already been stored in the ocean pre-condition before April, and the influence of the ocean pre-condition on MWP genesis indirectly reflects the influence of surface heat fluxes during pre-monsoon. As a result, we will need a longer simulation to reach this conclusion. Therefore, following the Reviewer's suggestion, we have removed the statement 'This contradicts previous studies, such as Kurian and Vinayachandran (2007), which suggested that MWP development in May was independent of the pre-April ocean conditions.'*

♦ *The ocean heat budget allows us to understand the impact of multiple factors (net surface heat flux, horizontal advection, vertical process, etc.) on variations in mixed layer temperature tendency. The mixed layer heat budget from May 1st to June 10th showed that the temperature tendency in the MWP core region was mostly determined by net surface heat flux. Furthermore, the net surface heat flux, together with vertical processes, caused MWP's dissipation.*

*The Reviewer advised showing the mixed layer heat budget from April to June. We can understand the Reviewer's concern regarding the fewer positive temperature tendency days*

*from May to June. However, if we show the mixed layer heat budget from April to June, the importance of different processes on the MWP development and dissipation does not change. Besides, in this whole study, we have left the April month for spin-up time and analyzed the MWP and associated processes from May to June (we have mentioned it clearly in line 167-168 in the revised manuscript). Thus, including the spin-up time analysis only in the mixed-layer heat budget could break the study's consistency. Subsequently, we have restricted ourselves from showing April's mixed layer heat budget in the main manuscript. As the discussion section is open in EGU Ocean Science, we have shown the figure here and have mentioned this in the Fig. 11 caption of the main manuscript. Interested readers can find it.*

**Mixed Layer Heat Budget**

**Fig R 1 Area averaged (72- 76ºE and 7-13ºN) mixed layer heat budget for three control ((a) 2018 control experiment, (b) 2013 control experiment, and (c) 2016 control experiment) and four sensitivity experiments ((d) $S_{ocean2013}$, (e) $S_{ocean2016}$, (f)$S_{atmos2013}$, and (g) $S_{atmos2016}$). In the sensitivity experiments, the oceanic and atmospheric conditions have been changed to various years; thus, only the day and month are kept on the x-axis (d to g).**

2. I found the section 3.3.2 on potential TKE production weak, mainly because the figures were too difficult to interpret. A probable improvement would be to provide timeseries for box averages of the terms in Fig. 13-15, for the region boxed in Figs 2c,f,i).

**Reply:**

*We sincerely thank the Reviewer for this comment. The $P_{TKE}$ caused by haline, or thermal buoyancy flux, has less magnitude than the $P_{TKE}$ caused by wind. Due to this reason, we have focused on the $P_{TKE}$ caused by wind. Thus, the $P_{TKE}$ caused by the wind for all the experiments is combined and kept as a single figure in the revised main manuscript (Fig. 14), and the $P_{TKE}$ caused by haline and thermal buoyancy flux is moved to the supplementary section (Fig. S11 and S12). The spatial pattern of wind stirring is very important as it gives us information about the weak wind zone in the southeastern Arabian Sea as the spatial extent of the MWP expands within this weak wind zone.*

*Following the Reviewer's advice, we have shown the line plot of the $P_{TKE}$ caused by wind stirring, haline, and thermal buoyancy flux over the MWP core (Fig. 13 in the main manuscript) along with the $P_{TKE}$ caused by wind stirring (Fig. 14 in the revised manuscript). This certainly helped to understand the relative importance of wind stirring, haline and thermal buoyancy flux on the $P_{TKE}$.*

3. Map figures, e.g. Fig. 2, 3,4, 6, 7, 9, 10. The figures show a vast area of the Indian Ocean, and it can be hard to see the region of interest. I understand that you sometimes want to display large-scale features, but for some figures you can show a smaller area, e.g. north of 5°N and east of 60° For example, in Fig. 4 it is hard to see coastal currents, and a smaller display region focused on the coasts would be better. For Figs 13-15, replace with timeseries plots as in Major Comment 2.

**Reply:**

*We wholeheartedly thank the Reviewer for this suggestion. The idea behind showing the vast area was to show the large-scale processes, especially the wind flow, as shown in Fig. 6. Besides, we aimed to show the model validation over the whole model domain in Fig. 2 and 3. However, we agree with the Reviewer that the coastal currents are not properly visible in Fig. 4. Hence, we have shown the currents near the west coast of India following the Reviewer's suggestion. We have also marked the domain of interest (the MWP core) in all the figures, which showed a vast area. Fig. 13 in the revised main manuscript shows the time series of the $P_{TKE.}$*

**Line-by-line points**

1. Line 13-14. See Major Comment 1.

**Reply:**

*We have addressed the Reviewer's concern regarding the simulation time and the longer period of the mixed layer heat budget in the major comment one, where we have shown the temperature tendency from April to June. Please have a look at the comments of the major comment 1.*

2. Line 151. An earlier reference is Stevenson and Niiler 1983 - https://doi.org/10.1175/1520-0485(1983)013%3C1894:UOHBDT%3E2.0.CO;2

**Reply:**

*We thank the Reviewer for this suggestion. The above citation is added. Please look at the lines 180-181.*

3. Lines 169-170. The expression in Han et al. 2001 does not include the von Karman constant, and I could not find this expression in Rao et al. 2002. Please check, I may have missed it.

**Reply:**

*We have used the Von Kärmän constant from Rao et al. (2002). We request you to look closely at the Section 3 ("Causes for the Absence of Cool SST Anomalies in the Bay of Bengal") of Rao et al. (2002) for the value of Von Kärmän constant (k = 0.42).*

4. Lines 208-214. A bit more detail on the analysis – were you looking at daily or monthly variability? How long were the data records? Do you remove any seasonal cycle or trend?

**Reply:**

*We thank the Reviewer for this comment. We have looked at the daily variability at the point location nearest to the AD10 buoy location. We used the AD10 for model validation purposes. Thus, we have kept the original data and have not removed any seasonality or trend. In the main manuscript, we have added a few details regarding the comparison between the AD10 and the model's ability to simulate the vertical temperature and salinity profile. Kindly have a look at lines 230-247.*

5. Line 189-190. Please put the time period also in the caption of Fig. 2.

**Reply:**

*We have added the time period in the captions of Fig.2, 3, and Fig. 4. Thank you for this suggestion.*

6. Lines 253-257. This process is hard to see in Fig. 6. Please explain more or delete.

**Reply:**

*The manuscript is already quite lengthy, and the south-eastward transport of salinity is not very relevant to the objective of this study. Hence, we have removed this information from the manuscript. Thank you for this advice.*

7. Line 281 "enhance"->"allow".

**Reply:**

*We have incorporated this comment. However, following the Reviewer 1's suggestion, we have moved it to the section 3.3.1. Please have a look at the lines 358-359.*

8. Lines 291-300. See Major Comment 1.

**Reply:**

*We thank the Reviewer for this comment. This particular comment is addressed in detail in the major points 1.*

9. Line 303. At this point, remind readers that the focus (control) year is 2018, and sensitivity experiments impose non-2018 conditions.

**Reply:**

*We thank the Reviewer for this comment. The section 3.3 is restructured following the comments of both the Reviewers. In this process, we have included this information in section 3.3.1. We request the Reviewer to have a look at the lines 325-334.*

10. Figs 9-10. Please show 2018 control as an additional row for comparison.

**Reply:**

*We thank the Reviewer for this insightful comment. We have added the 2018 control experiment in an additional row in Fig. 9 and 10.*

11. Section 3.3.2 See Major Comment 2.

**Reply:**

 *We thank the Reviewer for this comment. We have added the time series to this section. Please see Fig. 13.*

12. Lines 423-424. See Major Comment 1.

**Reply:**

*Following the Reviewer's Major Comment 1, we have removed this information in the updated manuscript. Thank you for the advice.*

**Reference:**

*Akhil, V. P., Lengaigne, M., Krishnamohan, K. S., Keerthi, M. G., & Vialard, J. (2023). Southeastern Arabian Sea Salinity variability: mechanisms and influence on surface temperature. Climate Dynamics, 61(7–8). https://doi.org/10.1007/s00382-023-06765-z*

*Durand, F., Shetye, S. R., Vialard, J., Shankar, D., Shenoi, S. S. C., Ethe, C., & Madec, G. (2004). Impact of temperature inversions on SST evolution in the South-Eastern Arabian Sea during the pre-summer monsoon season. Geophysical Research Letters, 31(1). https://doi.org/10.1029/2003GL018906*

*Gopalakrishna, V. V., Johnson, Z., Salgaonkar, G., Nisha, K., Rajan, C. K., & Rao, R. R. (2005). Observed variability of sea surface salinity and thermal inversions in the Lakshadweep Sea during contrast monsoons. Geophysical Research Letters, 32(18). https://doi.org/10.1029/2005GL023280*

*Hareesh Kumar, P. V., Joshi, M., Sanilkumar, K. V., Rao, A. D., Anand, P., Anil Kumar, K., & Prasada Rao, C. V. K. (2009). Growth and decay of the Arabian Sea mini warm pool during May 2000: Observations and simulations. Deep-Sea Research Part I: Oceanographic Research Papers, 56(4). https://doi.org/10.1016/j.dsr.2008.12.004*

*Kurian, J., & Vinayachandran, P. N. (2007). Mechanisms of formation of the Arabian Sea mini warm pool in a high-resolution Ocean General Circulation Model. Journal of Geophysical Research: Oceans, 112(5). https://doi.org/10.1029/2006JC003631*

*Masson, S., Luo, J. J., Madec, G., Vialard, J., Durand, F., Gualdi, S., Guilyardi, E., Behera, S., Delecluse, P., Navarra, A., & Yamagata, T. (2005). Impact of barrier layer on winter-spring*

variability of the southeastern Arabian Sea. *Geophysical Research Letters, 32(7).* *https://doi.org/10.1029/2004GL021980*

Mathew, S., Natesan, U., Latha, G., & Venkatesan, R. (2018). *Dynamics behind warming of the southeastern Arabian Sea and its interruption based on in situ measurements. Ocean Dynamics, 68(4–5). https://doi.org/10.1007/s10236-018-1130-3*

Nyadjro, E. S., Subrahmanyam, B., Murty, V. S. N., & Shriver, J. F. (2012). *The role of salinity on the dynamics of the Arabian Sea mini warm pool. Journal of Geophysical Research: Oceans, 117(9). https://doi.org/10.1029/2012JC007978*

Rao, S. A., Gopalakrishna, V. V., Shetye, S. R., & Yamagata, T. (2002). *Why were cool SST anomalies absent in the Bay of Bengal during the 1997 Indian Ocean Dipole event? Geophysical Research Letters, 29(11). https://doi.org/10.1029/2001GL014645*

---

## Referee Report (RR1)

**Review of "An evaluation of the Arabian Sea Warm Pool's advancement during its mature phased using a coupled atmosphere-ocean numerical model" by S. P. Lahiri, K. R. Prakash, and V. Pant, submitted to EGUsphere.**

The authors have carefully considered my comments and improved the manuscript. The paper is generally well written, except for minor points, some of which are noted below. However I still have one major comment, so that I give a Major Revision.

**Major points**

I have looked again at the heat budget results in Fig. 11 and your Response fig. R1.   In Fig. 11 there is hardly any actual warming (positive temperature tendency) in any of the panels. In the extended Response figure there is more sign of warming: in 2018 it is positive from early April towards end of April, in 2016 it is mostly positive in April except for a few days: while in 2013 two warming periods are separated by a ~ week long period of cooling.

Based on this, if you use heat budgets from May onwards, I think the sentence in the Conclusions "(Fig 11 ) revealed that the net surface heat flux is the primary driver of the MWP development" is hard to justify, as you do not actually show substantial warming (positive tendency) in the heat budget.  Even if surface heat flux contributes 0.1deg.C per day, it can be cancelled out by other processes so the net tendency is small (e.g. Fig. 11a, May 13-22).

Your results are better for looking at the dissipation process, where surface fluxes and vertical processes play a large role. If you want to relate it to the maturing phase, I think you have to include April, even though it is spin-up.

Alternatively, it might be useful to compare heat budgets in the mini-warm pool with points outside the warm pool. From the difference of these two locations, is more surface heat flux leading to more positive tendency in the warm pool?

I request that you re-write the sentence in the Conclusions to reflect the thoughts above.

**Minor points**

Line 228. The coastal currents are still very hard to distinguish. Perhaps zoom in (leave larger plots for supplemental) and plot arrows more frequently and change color scale.

Lines 444-445. Net surface heat flux is the primary driver behind dissipation in 2016, but in the other years vertical processes are also very important.

Lines 450-451. Vertical processes are influenced by the atmosphere (e.g. wind) and ocean (e.g. stratification, mixing) so I would not say it is all "atmospheric processes".

**Wording changes**

Some improvement in wording/language-style is required. I list below some examples, but please read to check for more.

Line 152. "boundaries were closed" – strictly the boundaries are over land anyway.

Line 152. Plural "models"

Line 161 "simulated the model" -> "ran the model for about 80 days"

Line 167. Reword to "separately on April 1$^{st}$ and run to June 20 each year"

Lines 187-188 "The MLD is the depth h where the following criterion is first met" ?

Line 195. Delete "represented in the"

Move lines 191-196 to after the sentence ending in "h is the mixed layer depth (MLD)." Then move lines 186-190 to the end of the sub-section.

Line 235 "anticipated"-> "simulated"

Line 276, figure 5 caption. Move "points at 50m… for 2013" to main text instead.

Line 330 "till" - > "to"

Line 335. "the 2013 atmospheric conditions replaced those from 2018"

Fig. 11 caption. Please remove the last sentence for the print version.

Line 404. "minimal" - > "opposite sign and smaller magnitude"

Fig. 13 use same vertical axis scale for all panels.

Caption and text. Move "multiplying 10^4" to the vertical axis label.

Line 417. "was unfurled over" is too fancy, I suggest "expanded to"

Lines 446-449 have confusing wording. Please rewrite.

Line 453. "rise of 41%" -> "decrease of 41%"? See line 393.

Line 457. "S_atmos2016"->"S_ocean2013" ? See line 388.

Line 458. "it's" to "it is" reads better.

Line 480-481. Delete sentence beginning "Given that …". Reword as "We hypothesize that the wind shadow zone and the corresponding increase…"

---

## Author Response (AR2)

**Author's Response to Reviewer's Comment**

**Manuscript No. -** egusphere-2024-2848

**Title: "**An evaluation of the Arabian Sea Mini Warm Pool's advancement during its mature phase using a coupled atmosphere-ocean numerical model"

**Comments to Reviewer - 1**

**Review of "An evaluation of the Arabian Sea Warm Pool's advancement during its mature phased using a coupled atmosphere-ocean numerical model" by S. P. Lahiri, K. R. Prakash, and V. Pant, submitted to EGUsphere.**

I want to recognize the effort the authors have made after the first round of reviews. The manuscript is noticeably improved. However, I do have some additional comments.

Following my original comment #4 and Major point 1 from the other reviewer regarding the short spin-up and how close to the time frame of interest the spin-up window is (particularly considering ocean memory). The authors should make a comment about this in the manuscript much like in their response to reviewer 2. In addition, the point Reviewer #2 made about the ocean pre-conditioning/memory is very important even if the paper focuses on the mature and dissipation phases. I believe these merits further clarification/justification in the manuscript.

Reply:

*We sincerely thank the Reviewer for the encouraging words. Your time and effort in thoroughly reviewing the manuscript is much appreciated. These comments and discussion have substantially contributed to enhance the scientific quality of the manuscript. Following your suggestions, we have edited the manuscript accordingly. Your comments have been addressed individually, as outlined below, and the corresponding revisions are incorporated into the manuscript. Your comments are presented in black, and our responses are provided in blue italic font. For comments containing multiple queries, we have addressed each point as bullet points for clarity.*

- *We agree with the reviewer that the spin up time is less in our configured numerical model. However, as we have already reported in our previous comments that in our reanalysis*

*feed coupled atmosphere ocean numerical model (the atmospheric and oceanic initial and boundary conditions are given from reanalysis data), one month spin up time is sufficient to solve the mixed layer processes. Besides, the computational cost increases exponentially with an increase in the spin up time. Thus, we have restricted ourselves to one month spin up time. However, following the comment 1 of the Reviewer 2, we now have shown the mixed layer heat budget from mid-April although it is within the spin up time. This further clarifies the impact of the net surface heat flux on the temperature tendency of the mini warm pool.*

- *One of the major findings of our study is that the ocean precondition plays a dominant role in the genesis of the mini warm pool. Thus, we are agreeing with the reviewer that the ocean memory (or ocean precondition) indeed plays an important role for the development of the mini warm pool. However, a substantial debate regarding the actual timeline of the ocean warming before the formation of the mini warm pool still persists within the scientific community (please have a look at the introduction especially lines 61 to 69). It is quite possible that a strong mini warm pool could start to develop from the antecedent year. However, the objective of the present study was to investigate the relative role of the atmosphere and the ocean-pre-condition on the mini warm pool's advancement during its mature phase. Since, we have concluded that the ocean pre-condition indeed impacts the mini warm pool formation, it becomes very important to study the timeline of the warming in the southeastern Arabian Sea before a strong mini warm pool formation. We have added this information in lines 475 to 480 in the conclusion section. Please have a look at it.*

Line 92: Please include Bonjean and Lagerloef (2002) for the OSCAR dataset.

Reply:

*We thank the reviewer for pointing this out. We have added the citation. Please have a look at the lines 93-94.*

In the response to comment 9 on my original review regarding SST biases when comparing the model to AVHRR, the authors say "However, most of these biases are within 1.5-2C." But in the revised manuscript the authors still state "The SST bias remained within 1C in all three experiments except for the northern Arabian Sea along the somalian coast." (line 218). In the SEAS region, Figure 2f shows a signal with a bias over 1oC.

Reply:

*We have missed to add the information. It is corrected in the revised manuscript. Please have a look at the lines 219. Thank you for pointing this out.*

The authors now include a link to the AD10 buoy data as part of the Data Availability Statement, but the link or some other reference should be included in Line 230 when the buoy is first mentioned. Figure S2 should include the box with the SEAS region of interest.

Reply:

- The link to the AD10 buoy data is added in the lines 94-97 in the Data and Methodology section (the buoy is mentioned here for the first time).
- The SEAS region is marked in the Fig. S2.

We thank the reviewer for pointing this out.

The validation in 2013 with the buoy is questionable (Figure S4). You are missing the top ~10 m in temperature and salinity. The salinity data from AD10 shows a large discrepancy near the data gap during the first third of the time series. Given how important the surface signals are for the purposes of the research, the lack of data in the top 10m and the odd signal in salinity leads me to question the robustness/usefulness of this validation within the context of the manuscript. Furthermore, the validation with the buoy data is a relatively long paragraph in the main text but all figures are relegated to the supplemental (except for the Taylor diagram that is only a couple of sentences.

Reply:

- We can understand the reviewer's concern regarding the model's performance in 2013. AD10 data has a substantial data gap within top 10 m.  Hence, we can't compare the vertical temperature and salinity within this depth. But below that the configured coupled model validates well for both temperature and salinity and the bias remains within 1°C for temperature and 1psu for salinity. Besides, the surface salinity and temperature are well represented in the numerical model when compared with the satellite data  in 2013 (Fig. 2 and 3). This gave us confidence to further use our configured model for 2013 sensitivity experiments.

- Following the reviewer's advice, we have removed few lines from the vertical temperature and salinity validation section in the main manuscript. We request the reviewer to please go through the lines 231-248.

Line 239: "The MWP does not extend beyond 50m…" Please include a reference for this.
Reply:
We have added a reference here. Thank you for pointing this out. Please have a look at the lines 239-240.

As previously requested by both reviewers, please include the region of interest in all figures and panels (not just the difference panels).
Reply:
We have added the region of interest in all the figures. Thank you for reminding this point.

Line 293-294: Figure 7 panels g, h, I do not seem to support the statement that salinity in the vicinity of the MWP was lower during its mature day but increased during its dissipation day. Figure 7i does not show an overall positive difference which would support the statement. Also, It is not clear to me how low salinity being slightly outside the core MWP area contradicts Kumar et al. 2019 as stated in lines 295-296. Maybe some rewriting with additional details could clarify this better.

Reply:

We sincerely thank the reviewer for this suggestion. We have rewritten the sentences. Please have a look at the lines 293-297.

Lines 446-447: I found these two sentences confusing, please be a bit more specific on what you are trying to say. For example in the second sentence, It is not clear what the authors are trying to say with regards the experiments.

Reply:

We have rewritten the statement here for better clarity. Please have a look at the lines 447-450.

Lines 449-450: I would suggest to rewrite this sentence also to be more precise. For example: "This net surface heat flux drives the dissipation of the MWP with contributions from the vertical processes. This result indicates that atmospheric processes control the MWP's …"
Reply:

We have rewritten the statement here for better clarity. Please have a look at the lines 447-450. Thank you for your suggestion.

**Author's Response to Reviewer's Comment**

**Manuscript No. -** egusphere-2024-2848

**Title: "**An evaluation of the Arabian Sea Mini Warm Pool's advancement during its mature phase using a coupled atmosphere-ocean numerical model"

**Comments to Reviewer - 2**

**Review of "An evaluation of the Arabian Sea Warm Pool's advancement during its mature phased using a coupled atmosphere-ocean numerical model" by S. P. Lahiri, K. R. Prakash, and V. Pant, submitted to EGUsphere.**

The authors have carefully considered my comments and improved the manuscript. The paper is generally, well written, except for minor points, some of which are noted below. However, I still have one major comment, so that I give a Major Revision.

**Major points**

I have looked again at the heat budget results in Fig. 11 and your Response fig. R1. In Fig. 11 there is hardly any actual warming (positive temperature tendency) in any of the panels. In the extended Response figure there is more sign of warming: in 2018 it is positive from early April towards end of April, in 2016 it is mostly positive in April except for a few days: while in 2013 two warming periods are separated by a ~ week long period of cooling.

Based on this, if you use heat budgets from May onwards, I think the sentence in the Conclusions "(Fig 11 ) revealed that the net surface heat flux is the primary driver of the MWP development" is hard to justify, as you do not actually show substantial warming (positive tendency) in the heat budget. Even if surface heat flux contributes 0.1deg.C per day, it can be cancelled out by other processes so the net tendency is small (e.g. Fig. 11a, May 13-22).

Your results are better for looking at the dissipation process, where surface fluxes and vertical processes play a large role. If you want to relate it to the maturing phase, I think you have to include April, even though it is spin-up.

Alternatively, it might be useful to compare heat budgets in the mini-warm pool with points

outside the warm pool. From the difference of these two locations, is more surface heat flux leading to more positive tendency in the warm pool?

I request that you re-write the sentence in the Conclusions to reflect the thoughts above.

Reply:

*We sincerely thank the Reviewer for the encouraging words. Your time and effort in thoroughly reviewing the manuscript is much appreciated. These comments and discussion have substantially contributed to enhance the scientific quality of the manuscript. Following your suggestions, we have edited the manuscript accordingly. Your comments have been addressed individually, as outlined below, and the corresponding revisions are incorporated into the manuscript. Your comments are presented in black, and our responses are provided in blue italic font. The major comments are in blue bold italic color font. For comments containing multiple queries, we have addressed each point as bullet points for clarity.*

- ***We agree with the statement that the influence of the net surface heat flux on the temperature tendency of the mini warm pool is much evident from April onwards. However, for the sake of consistency (as we have excluded the spin up time for the analysis in the whole manuscript), we did not intend to keep the longer timeseries of the mixed layer heat budget in the main manuscript. But, following your major comment, we have now replaced the mixed layer heat budget from May with the mixed layer heat budget from April (Fig. 11 in the revised manuscript). This certainly helps to understand the influence of the net surface heat flux on the MWP temperature tendency during the mature phase.***

- ***To better comprehend the influence of the net surface heat flux on the mini warm pool temperature tendency, we have compared the mixed layer heat budget in the mini warm pool (7-12°N & 72-77°E) (Fig. R1 a-c) and 5° west of the mini warm pool (7-12°N & 60-65°E) (Fig. R1 d-f). A substantial decrease in the temperature tendency is observed in the west of mini warm pool in all the three years. The impact of the net surface heat flux on the temperature tendency is also less here. However, in the vicinity of the mini warm pool (Fig. R1 a-c), the temperature tendency is more and so as the influence of the net surface heat***

*flux on the temperature tendency, indicating the importance of the net surface heat flux in the development of the mini warm pool during its mature phase.*

[Figure]

**Fig R 1 Area averaged mixed layer heat budget in the mini warm pool core region (72- 76ᵒE and 7-13ᵒN, i.e., the white box shown in Fig. 1) and 5ᵒ west of the mini warm pool region (72- 76ᵒE and 7-13ᵒN, i.e., the white box shown in Fig. 1) for three control ((a and d) 2018 control experiment, (b and e) 2013 control experiment, and (c and f) 2016 control experiment).**

*After keeping the mixed layer heat budget from April onwards, the influence of the net surface heat flux on the mini warm pool temperature tendency becomes more evident. Besides, it now supports our conclusion statement i.e., "(Fig 11 ) revealed that the net surface heat flux is the primary driver of the MWP development".*

**Minor points**

Line 228. The coastal currents are still very hard to distinguish. Perhaps zoom in (leave larger plots for supplemental) and plot arrows more frequently and change color scale.

Reply:

*We thank the reviewer for pointing out this. The coastal currents along the west coast of India*

*remains in the transition phase during the May and late June and it strengthens in July-August (Fig. R2). Thus, we are not seeing any prominent current along the west coast, indicating the model's capability in capturing the current precisely. However, we have now dense the arrow and changed the color scale limit. We hope that the currents are much more visible now.*

[Figure]

***Fig R 2** OSCAR surface current along the west coast of India during May-June(MJ) and July-August (JA) for the year 2018, 2013 and 2016.*

Lines 444-445. Net surface heat flux is the primary driver behind dissipation in 2016, but in the other years vertical processes are also very important.

Reply:

*We have rewritten this sentence as "Thus, the net surface heat flux along with vertical processes emerges as the primary driver behind the dissipation of the MWP (Fig. 11)." Please have a look at the lines 445-446. Thank you for pointing this out.*

Lines 450-451. Vertical processes are influenced by the atmosphere (e.g. wind) and ocean (e.g. stratification, mixing) so I would not say it is all "atmospheric processes".

Reply:

*We thank the reviewer for bringing this point to our attention.*

*The mini warm pool extends till the mixed layer depth, and it expands following a weak wind zone. However, once the southwesterly monsoon wind strengthens it causes strong vertical mixing that negatively impact the mini warm pool's temperature and leads to its dissipation.*

*Besides, the moisture rich southwesterly wind causes cloud formation which restricts the incoming shortwave radiation and at the same time the loss of latent heat flux due to evaporation. This leads to the negative influence of the wind on the net surface heat flux which eventually affect the temperature tendency of the mini warm pool.*

*Thus, once the ocean pre-condition laid a favorable condition, the mature to the dissipation day of the mini warm pool is driven by the wind. We have modified this in the lines 447-450.*

**Wording Changes:**

1. Line 152. "boundaries were closed" – strictly the boundaries are over land anyway.
   Reply:
   *We have re-written the sentence as "Northern and western boundaries were closed in the ROMS model." See lines 153-154.*

2. Line 152. Plural "models"
   Reply:
   *This is rectified in the manuscript. Please have a look at the line 154.*

3. Line 161 "simulated the model" -> "ran the model for about 80 days"
   Reply:
   *This is modified in the lines 163-164.*

4. Line 167. Reword to "separately on April 1st and run to June 20 each year"

Reply:

*The line is rewritten. Please check line 169.*

5. Lines 187-188 "The MLD is the depth h where the following criterion is first met" ?

Reply:

*We are not sure what the reviewer wants to mean here. If this is about the mixed layer depth calculation, then we request the reviewer to see the lines 194-198.*

6. Line 195. Delete "represented in the"

Reply:

 *We have reframed this in line 192-193.*

7. Move lines 191-196  to after the sentence  ending in "h is the mixed layer depth (MLD)."
   Then move lines 186-190 to the end of the sub-section.

Reply:

*We have reframed this section as suggested. Please have a look at the lines 186-195.*

8. Line 235 "anticipated"-> "simulated"

Reply:

*We have replaced the word. Please have a look at the lines 235-236.*

9. Line 276, figure 5 caption. Move "points at 50m… for 2013" to main text instead.

Reply:

*We thank the reviewer for this comment. We have added this information in the main text line no 245.*

10. Line 330 "till" - > "to"

Reply:

*This is incorporated in the text. Please have a look at the lines 329-331.*

11. Line 335.  "the 2013 atmospheric conditions replaced those from 2018"

Reply:

*The lines are reformed in 335-336.*

12. Fig. 11 caption. Please remove the last sentence for the print version.

Reply:

*We have removed this in the revised manuscript.*

13. Line 404. "minimal" - > "opposite sign and smaller magnitude"

Reply:

*Please have a look at 404-405 for the updated version.*

14. Fig. 13 use same vertical axis scale for all panels. Move "multiplying 10^4" to the vertical axis label.

Reply:

*The vertical axis is kept in all the panels and "multiplying 10^4" is moved to the vertical axis label.*

15. Line 417. "was unfurled over" is too fancy, I suggest "expand to"

Reply:

*We have incorporated this in the lines 418.*

16. Lines 446-449 have confusing wording. Please rewrite.

Reply:

*These sentences are re-written. Please have a look at the lines 447-450. We thank the reviewer for this suggestion.*

17. Line 453. "rise of 41%" -> "decrease of 41%"? See line 393.

Reply:

*We have restructured the whole section for clarity. Please have a look at the lines 447-458.*

18. Line 457. "S_atmos2016"->"S_ocean2013" ? See line 388.

Reply:

*We have restructured the whole section for clarity. Please have a look at the lines 447-458.*

19. Line 458. "it's" to "it is" reads better.

Reply:

20. *We have restructured the whole section for clarity. Please have a look at the lines 447-458.*

21. Line 480-481. Delete sentence beginning "Given that …". Reword as "We hypothesize that the wind shadow zone and the corresponding increase…"

Reply:

*We have modified this sentence. Please have a look at the lines 481-483.*